



# Evaluating Sentinel-5P TROPOMI tropospheric NO₂ column densities with airborne and Pandora spectrometers near New York City and Long Island Sound

Laura M. Judd[1], Jassim A. Al-Saadi[1], James J. Szykman[2], Lukas C. Valin[2], Scott J. Janz[3], Matthew G. Kowalewski[3,4], Henk J. Eskes[5], J. Pepijn Veefkind[5,6], Alexander Cede[7], Moritz Mueller[7], Manuel Gebetsberger[7], Robert Swap[3], R. Bradley Pierce[8], Caroline R. Nowlan[9], Gonzalo González Abad[9], Amin Nehrir[1], David Williams[2]

[1]NASA Langley Research Center, Hampton, VA, 23681, United States
[2]United States Environmental Protection Agency Office of Research and Development, Triangle Research Park, NC, 27709, United States
[3]NASA Goddard Space Flight Center, Greenbelt, MD, 20771, United States
[4]Universities Space Research Association, Columbia, MD, 21046, United States
[5]Royal Netherlands Meteorological Institute (KNMI), De Bilt, Netherlands
[6]Delft University of Technology, Department of Geoscience and Remote Sensing, Delft, Netherlands
[7]LuftBlick, Kreith, Austria
[8] University of Wisconsin-Madison Space Science and Engineering Center, Madison, WI, 53706, United States
[9]Harvard-Smithsonian Center for Astrophysics Cambridge, MA, 02138

*Correspondence to*: Laura M. Judd (laura.m.judd@nasa.gov)

**Abstract.** Abundant NO₂ column measurements from airborne and ground-based Pandora spectrometers were collected as part of the 2018 Long Island Sound Tropospheric Ozone Study (LISTOS) in the New York City/Long Island Sound region and coincided with early measurements from the Sentinel-5P TROPOMI instrument. Both airborne- and ground-based measurements are used to evaluate the TROPOspheric Monitoring Instrument (TROPOMI) NO₂ Tropospheric Vertical Column (TrVC) product v1.2 in this region, which has high spatial and temporal heterogeneity in NO₂. First, airborne and Pandora TrVCs are compared to evaluate the uncertainty of the airborne TrVC and establish the spatial representativeness of the Pandora observations. The 171 coincidences between Pandora and airborne TrVCs are found to be highly correlated (r²=0.92 and slope of 1.03) with the biggest individual differences being associated with high temporal and/or spatial variability. These reference measurements (Pandora and airborne) are complementary with respect to temporal coverage and spatial representivity. Pandora spectrometers can provide continuous long-term measurements but may lack areal representivity when operated in direct-sun mode. Airborne spectrometers are typically only deployed for short periods of time, but their observations are more spatially representative of the satellite measurements with the added capability of retrieving at subpixel resolutions of 250 m × 250 m over the entire TROPOMI pixels they overfly. Thus, airborne data are more correlated with TROPOMI measurements (r²=0.96) than Pandora measurements are with TROPOMI (r²=0.84). The largest outliers between TROPOMI and the reference measurements are caused by errors in the TROPOMI retrieval of cloud pressure impacting the calculation of tropospheric air mass factors in cloud-free scenes. This factor causes a high bias in TROPOMI



TrVCs of 4-11%. Excluding these cloud-impacted points, TROPOMI has an overall low bias of 19-33% during the LISTOS timeframe of June-September 2018. Part of this low bias is caused by coarse a priori profile input from TM5-MP model; replacing these profiles with those from a 12km NAMCMAQ analysis results in a 12-14% increase in the TrVCs. Even with this improvement, the TROPOMI-NAMCMAQ TrVCs have a 7-19% low bias, indicating needed improvement in a priori

assumptions in the air mass factor calculation. Future work should explore additional impacts of a priori inputs to further assess the remaining low biases in TROPOMI using these datasets.

# 1 Introduction

Nitrogen dioxide ($NO_2$) is an air pollutant emitted naturally through soil emissions and lightning, and anthropogenically as a combustion product from sources such as mobile vehicles, powerplants, and industrial processes. $NO_2$ is harmful to human

health (e.g., Fischer et al., 2015; Anenberg et al., 2018) both directly and through its role in the production of near-surface ozone and particulate matter making it a criteria air pollutant monitored and regulated by the Clean Air Act (https://www.epa.gov/clean-air-act-overview: last accessed 18 April 2020). Due to its short lifetime of a few hours as a component of NOx (NO + $NO_2$) (Liang et al., 1998; Beirle et al., 2011; Liu et al., 2016), the spatial distribution of $NO_2$ near anthropogenic emission sources is highly heterogeneous with complex patterns that are hard to characterize from sparse

networks of ground-based monitors.

The TROPOspheric Monitoring Instrument (TROPOMI) on board the Copernicus Sentinel-5 Precursor (S5P) satellite currently measures column densities of $NO_2$ globally at unprecedented spatial resolution making it an important tool for studying and monitoring urban air pollution. TROPOMI continues a long legacy of UltraViolet-VISible (UV-VIS) backscatter measurements from satellites observing trace gas column densities related to air quality (González Abad et al., 2019). Global

$NO_2$ measurements have heritage from the Global Ozone Monitoring Experiment (GOME; Burrows et al., 1999), SCanning Imaging Absorption SpectroMeter for Atmospheric CHartographY (SCIAMACHY; Bovensmann et al., 1999), GOME-2 (Callies et al., 2000; Behrens et al., 2018), Ozone Monitoring Instrument (OMI; Levelt et al., 2006; Levelt et al., 2018), Ozone Mapping and Profiling Suite (OMPS; Yang et al., 2014), and as of October 2017, TROPOMI (Veefkind et al., 2012) aboard S5P. Over the last couple decades, the spatial and temporal resolution of these satellite $NO_2$ products have improved with the

first daily global coverage achieved by OMI launched in 2004 and with TROPOMI achieving a spatial resolution an order of magnitude finer (currently approximately 3.5 km × 5.5 km at nadir) than the still-operating OMI and OMPS instruments.

The use of the TROPOMI tropospheric $NO_2$ products for applications such as evaluating emissions inventories and distinguishing point sources has already been documented in recent literature. Goldberg et al. (2019) used data from the first year of TROPOMI operation to evaluate top-down NOx emissions over three major U.S. cities and two large powerplants.

Complementary studies also pinpointed emissions from large point sources (Beirle et al., 2019) and even showed that emissions in Paris, France, have not decreased as expected since 2012 (Lorente et al., 2019). Griffin et al. (2018) found that the improved



spatial resolution of TROPOMI was able to distinguish $NO_2$ plumes from individual sources near the Canadian Oil Sands, which was not possible with the coarser measurements from OMI.

To enhance the integrity of using TROPOMI data in research and applications, each product requires systematic evaluation and validation. Validation activities include evaluating the data products under polluted and clean scenes using reference measurements from satellite, airborne, and ground-based instrumentation (van Geffen et al., 2019). Routine TROPOMI $NO_2$ validation reports are produced regularly and documented at http://mpc-vdaf.tropomi.eu/ (last accessed: 30 March 2020). Additional in-depth studies in recent literature have been mostly confined to ground-based column measurements from MAX-DOAS and/or direct-sun column measurements (e.g., from Pandora spectrometers) (e.g., Griffin et al., 2018, Zhao et al., 2019, Ialongo et al., 2020, Wang et al., 2020). These types of measurements have been used in the past to evaluate the OMI Tropospheric Vertical Column (TrVC) product, though this was shown to be challenging in polluted areas as spatial variability in $NO_2$ can result in sampling mismatches between the small spatial scale measurements from the ground-based spectrometers and the $> 300$ km$^2$ pixels from OMI (Lamsal et al., 2014; Reed et al., 2015; Goldberg et al., 2017; Judd et al., 2019). Initial results of TROPOMI $NO_2$ product validation with Pandora spectrometer direct-sun measurements show more encouraging results with higher levels of correlation than OMI evaluations (OMI examples found in Goldberg et al., 2017 and Judd et al., 2019; TROPOMI examples found in Griffin et al., 2018, Zhao et al., 2019, Ialongo et al., 2020, and this work).

In addition to ground-based column measurements, airborne column mapping datasets have been identified as valuable for TROPOMI TrVC validation efforts (van Geffen et al., 2019). Airborne spectrometers have the capability to map at much finer spatial resolutions than current satellite-based observations; for example, those used in this study have a spatial resolution of approximately 250 m × 250 m. Airborne spectrometers have been used to visualize high spatiotemporal variations in $NO_2$ over select areas in Europe, North America, Africa, and Asia (Popp et al., 2012; Schönhardt et al., 2015; Lawrence et al., 2015; Nowlan et al., 2016, 2018; Lamsal et al., 2017; Meier et al., 2017; Tack et al., 2017, 2019, Broccardo et al., 2018; Judd et al., 2018, 2019) and have even contributed toward evaluating emissions inventories and ozone production sensitivity (Schönhardt et al., 2015; Souri et al., 2018; Souri et al., 2020). Measurements from airborne spectrometers have also been compared to the OMI $NO_2$ products. Broccardo et al. (2018) found that agreement between the airborne mapper, iDOAS, and OMI improves with distance away from large emission source regions. Lamsal et al. (2017) discovered moderate correlation during a small subset of comparisons between the Airborne Compact Atmospheric Mapper (ACAM) and OMI over the Maryland region in 2011, though large differences were found for instances with insufficient sampling by the airborne mapper in areas subject to spatial heterogeneity of $NO_2$. The large pixels from OMI are difficult to completely sample with airborne spectrometer observations; however, with the improved spatial resolution of TROPOMI, representative sampling by airborne spectrometers is less of a concern as will be demonstrated in this work.

In this study, we use data from two NASA airborne spectrometers and nine ground-based (Pandora) spectrometers to evaluate the S5P TROPOMI $NO_2$ TrVC v1.2 product over New York City (NYC) and Long Island Sound during the summer 2018 Long Island Sound Tropospheric Ozone Study (LISTOS). The intercomparisons between the three independent datasets help bound $NO_2$ product uncertainties due to spatial and temporal variability and a priori assumptions within the retrievals.



Section 2 introduces LISTOS and each NO$_2$ dataset: S5P TROPOMI, the airborne spectrometers, and Pandora spectrometer, along with details on methodology. Section 3 evaluates the airborne spectrometer retrieval using Pandora measurements. Section 4 presents comparisons of TROPOMI NO$_2$ columns to the airborne spectrometer observations during LISTOS. Section 5 compares TROPOMI NO$_2$ TrVCs to Pandora spectrometer data for the LISTOS timeframe as well as expanded

through winter 2019. Throughout these sections causes for bias in the TROPOMI product based on the a priori profile and cloud assumptions are discussed. Section 6 summarizes TROPOMI NO$_2$ TrVC performance in the NYC region and Sect. 7 presents concluding remarks. Together these results demonstrate TROPOMI's capability for observing the spatial distribution of NO$_2$ in heterogeneous environments and demonstrate approaches for resolving apparent differences associated with linking observations from different measurement strategies.

**2 Data and Methods**

Data in this study were acquired across the NYC and Long Island Sound region in the United States as part of the Long Island Sound Tropospheric Ozone Study (LISTOS: https://www.nescaum.org/documents/listos; https://www-air.larc.nasa.gov/missions/listos/index.html : last accessed 18 April 2020). LISTOS was a multi-organizational collaborative air quality study focused on understanding the sources and temporal emission profiles of the ozone precursors, nitrogen oxides

(NOx) and volatile organic compounds (VOCs), across the NYC metropolitan area and ozone formation and transport in this coastal region. Measurements conducted include in situ and remotely sensed air quality and meteorology measurements from satellites, aircraft, and ground sites as well as the integration of the measurements with air quality models. This urban to sub-urban coastal area is a diverse region for validating satellite products due to the heterogeneous patterns in pollution as well as varying environmental factors such as surface reflectivity. In this study, we consider measurements from the LISTOS

timeframe to be from late June through September 2018, though some measurements extended before and after this time period.

**2.1 S5P TROPOMI**

Sentinel-5 Precursor (S5P) was launched October 2017 into a sun-synchronous low Earth orbit with a 13:30 local equator crossing time. S5P carries a single instrument, TROPOMI, which consists of a hyperspectral spectrometer observing eight

bands spanning the ultraviolet (UV), visible (VIS), near infrared, and shortwave infrared portions of the electromagnetic spectrum (Veefkind et al., 2012). The S5P orbit combined with the wide TROPOMI swath width of 2600 km provide observations between approximately 17:00-19:00 UTC (13:00-15:00 EDT) over the New York City and Long Island Sound region, capturing the early afternoon spatial distribution of trace gas columns including CO (Borsdorff et al., 2018), HCHO (De Smedt et al., 2018), CH$_4$ (Hu et al., 2017), NO$_2$ (van Geffen et al., 2019 & 2020), SO$_2$ (Theys et al., 2017), and O$_3$ (Garane

et al., 2019).





In this work, the TROPOMI v1.2 NO$_2$ TrVC product is evaluated with airborne and ground-based column density measurements from 25 June 2018 - 19 March 2019 over the LISTOS domain. The retrieval is built on the heritage of the Ozone Monitoring Instrument DOMINO product (Boersma et al., 2011) including developments from the QA4ECV project (Boersma et al., 2018; van Geffen et al., 2019; http://www.qa4ecv.eu/: last accessed 18 April 2020). NO$_2$ total slant columns

are retrieved via the Differential Optical Absorption Spectroscopy (DOAS; Platt and Stutz, 2008) method in the visible window of 405-465 nm. Following the spectral fit, the slant columns are separated into their stratospheric and tropospheric components. The stratospheric column is estimated by assimilating the total columns in the TM5-MP model. The remaining tropospheric slant columns are converted into vertical columns through the calculation and application of air mass factors (AMFs; Palmer et al., 2001). A priori inputs for the tropospheric NO$_2$ AMF calculations include viewing and solar geometry, surface pressure

and NO$_2$ profile shape from the 1° × 1° TM5-MP model (Williams et al. 2017), 0.5° × 0.5° surface albedo climatology built upon 5 years of OMI data (Kleipool et al. 2008), and the FRESCO-S cloud fraction and cloud height (Loyola et al., 2018) (Table 1).

TROPOMI data during the time period of this analysis have a nadir spatial resolution of 3.5 km × 7 km, with pixel areas ranging from 32.5 - 129.5 km$^2$. Beginning on 6 August 2019, the nadir spatial resolution of the TROPOMI NO$_2$ product

is refined to 3.5 km × 5.5 km (Ludewig et al., 2020). TROPOMI is capable of observing pollution at a spatial resolution a factor of 10 times more refined than its predecessor satellite sensor, OMI (Levelt et al., 2006; Levelt et al., 2018).

Only TROPOMI data with qa_value = 1 are considered in this analysis, which removes pixels influenced by issues such as sun glint, missing retrieval information, or cloud radiative fractions (CRF) above 50% (van Geffen et al., 2019, Eskes et al., 2019). We note that qa_values down to 0.75 are deemed acceptable for most data uses but 2% or less of the TROPOMI

data in this work had qa_values between 0.75 and 1 and do not affect the results. This work also makes use of the averaging kernel and pressure profiles used in the retrieval to explore the impact of different NO$_2$ profile shapes within the air mass factor calculation and explores sensitivity of the results to cloud retrievals during clear-sky scenes.

Figure 1 shows the annual average of NO$_2$ TrVCs observed over the LISTOS region from April 2018-March 2019, depicting peak NO$_2$ in the domain of over $10\times10^{15}$ molecules cm$^{-2}$ over much of New York City. The largest value is over the

southern tip of Manhattan Island at a magnitude of $12\times10^{15}$ molecules cm$^{-2}$. The spatial distribution and dynamic range of NO$_2$ varies widely day-to-day over this region due to variable meteorology, emissions, and the lifetime of NO$_2$, as shown through examples in this analysis.

## 2.2 Airborne Spectrometers

Two airborne UV-VIS mapping spectrometers are used in this study: Geostationary Trace gas and Aerosol Sensor Optimization

(GeoTASO) and GEO-CAPE Airborne Simulator (GCAS). GeoTASO and GCAS are very similar instruments but differ in characteristics such as their size, weight, wavelength range, and sensitivity. Specific details about these two instruments can be found in Leitch et al. (2014), Kowalewski and Janz (2014), Nowlan et al. (2016), and Nowlan et al. (2018) with a brief



summary in Table 2. The two instruments have very similar performance with respect to the NO$_2$ retrieval. Due to varying aircraft availability during LISTOS, these instruments were flown either interchangeably or together during 16 flight days between 18 June 2018 – 19 October 2018.   Only flights from 25 June – 6 September (13 flight days) are considered in this analysis due to availability of the high-resolution model data used to provide the a priori NO$_2$ profile shapes in the full vertical column retrieval (Table 1).  GeoTASO was flown on the NASA LaRC HU-25 Falcon through June 30th and GCAS was flown on the NASA LaRC B200 from July through September. The HU-25 Falcon is a faster aircraft capable of mapping approximately a 50% larger area per flight than the B200. This capability enabled us to also conduct measurements for the second Ozone Water-Land Environmental Transition Study domain (OWLETS2: https://www-air.larc.nasa.gov/missions/owlets/index.html: last accessed 7 January 2020) during June flights over Baltimore, Maryland in the early morning and late afternoon hours (outside the S5P overpass window). The NASA LaRC B200 has two nadir-viewing remote sensing portals, allowing installation of a second instrument along with GCAS.  The second instrument from July through September was the High Altitude Lidar Observatory (HALO: Nehrir et al. 2019) providing co-located measurements of nadir profiles of aerosols and methane. This analysis uses HALO aerosol optical thickness (AOT) retrievals at 532 nm to discuss aerosol conditions qualitatively. GeoTASO was the second instrument for flights in October, allowing for direct comparison of GCAS and GeoTASO retrievals, however these flights did not coincide with any clear-sky TROPOMI overpasses.

Figure 1 shows the two basic "raster" patterns that were flown by the NASA aircraft to create gapless maps of the high spatial resolution spectra from which NO$_2$ TrVCs are retrieved.  Both airborne instruments have a swath width of approximately 7 km at the nominal flight altitude of 9 km (pressure altitude of 28,000 ft), thus flight lines are spaced slightly over 6 km apart to ensure overlap between adjacent swaths. Table 3 includes a summary of all flights considered in this study along with cloud conditions, number of coincidences with Pandora and TROPOMI (assuming coincidence criteria discussed in Sect. 2.4 and throughout this manuscript), and raster type.  All flight days included two flights lasting approximately 4-5 hours each (morning and afternoon). The small raster (white lines in Fig. 1) could be accomplished 2 times in one flight (4 times per day), repeatedly measuring the same area to observe the temporal variation throughout the day.  The large raster (black lines in Fig. 1) could only be flown once per flight (twice per day) and was meant to capture a more regional view of the spatial distribution of NO$_2$ on days with expected air pollution over Long Island Sound and the surrounding communities.

The NO$_2$ retrieval algorithm is identical for GCAS and GeoTASO.  The retrieval process is summarized here with additional detail in Judd et al. (2019). NO$_2$ differential slant columns are retrieved at an approximate spatial resolution of 250 m × 250 m in the spectral fitting window of 425-460 nm relative to an in-flight measured reference spectra using the open-source DOAS computing software, QDOAS (http://uv-vis.aeronomie.be/software/QDOAS/; last accessed 18 April 2020). Reference spectra were collected over areas with low and homogeneous NO$_2$ absorption over a 4-5-minute time period using nadir observations for each of the 30 across-track positions. Three separate references were collected during the LISTOS campaign: June 30[th] for all GeoTASO flights, July 2[nd] for the GCAS flights for this day only (due to unique instrument conditions), and August 5[th] for the rest of the GCAS flights as the instrument conditions were stable for the rest of the flight



period. All reference spectra were co-located with total column NO$_2$ measurements from Pandora spectrometers: 5.6×10$^{15}$ molecules cm$^{-2}$ at MadisonCT on June 30$^{th}$, 5.7×10$^{15}$ molecules cm$^{-2}$ at MadisonCT on July 2$^{nd}$, and 6.2×10$^{15}$ molecules cm$^{-2}$ at WestportCT on August 5$^{th}$, with values estimated to be over 50% stratospheric.

Fitted trace-gas absorption cross sections in the slant column spectral fit include NO$_2$ (Vandaele et al., 1998), O$_4$ (Thalman and Volkamer, 2013), water vapor (Rothman et al., 2009), CHOCHO (Volkamer et al., 2005), Ring spectrum (Chance and Kurucz, 2010), and a fifth-order polynomial. Average ± standard deviation spectral fitting uncertainties for the NO$_2$ slant columns during cloud-free scenes at cruising altitude for GeoTASO are 1.6×10$^{15}$ ± 0.3×10$^{15}$ molecules cm$^{-2}$ and for GCAS are 0.8×10$^{15}$ ± 0.1×10$^{15}$ molecules cm$^{-2}$. The differences in uncertainty between spectral fits are likely due to a minor
amount of under-sampling of the GeoTASO slit function, which has a slightly flattened top hat shape compared to the more purely Gaussian shape exhibited by GCAS.

For slant to vertical column conversion, air mass factors (AMFs) are calculated using the Smithsonian Astrophysical Observatory AMF Tool (Nowlan et al., 2016 & 2018), which packages the VLIDORT radiative transfer model (Spurr, 2006) for calculating scattering weights based on user-inputs of viewing and solar geometries, a priori assumptions about surface
reflectivity with Bidirectional Reflectance Distribution Function (BRDF) kernels, and meteorological and trace gas vertical profiles. AMFs are then calculated following the methodology of Palmer et al. (2001) as the integrated product of scattering weights and shape factor (e.g., Nowlan et al., 2016; Lamsal et al., 2017; Judd et al., 2019).

Table 1 compares a priori assumptions used for TROPOMI and airborne AMF calculations. For both retrievals, the spatial resolution of the a priori assumptions are coarser than those of the observations, but a priori assumptions for airborne
observations are at a finer resolution than those for TROPOMI. Airborne a priori NO$_2$ vertical profile shapes are obtained for the troposphere from hourly output from a parallel developmental simulation of the North American Model–Community Multiscale Air Quality (NAMCMAQ) model from the National Air Quality Forecasting Capability (NAQFC; Stajner et al., 2011) and stratospheric NO$_2$ climatology developed using the PRATMO Photochemical Box Model (Prather, 1992; McLinden et al., 2000; Nowlan et al., 2016). The stratospheric column is bias corrected daily using TROPOMI NO$_2$ stratospheric vertical
columns by calculating the average offset between the two datasets over the LISTOS domain for each day (ranging from 5x10$^{13}$ to 6x10$^{14}$ molecules cm$^{-2}$). This analysis only focuses on the below aircraft portion of the NO$_2$ columns from the aircraft, which is henceforth referred to as tropospheric vertical columns or TrVCs.

Surface reflectance over land is represented in the AMF tool input files with the isometric, geometric, and volumetric BRDF kernels given by the MODIS MCD43A1 product at 500m resolution at 470 nm averaged over the time period of the
LISTOS campaign (Lucht et al., 2000; Schaaf and Wang, 2015). Input over water includes only the isometric BRDF kernel, limited to a minimum of 3% Lambertian reflectivity (similar to Nowlan et al., 2016), as well as an added Cox-Monk kernel (derived through references from Cox and Monk, 1954; Nakajima and Tanaka, 1983; Gordon and Wang, 1992; Spurr 2014; and wind speed from the lowest layer of the NAM-CMAQ model and viewing and solar geometry). The brighter areas where



the isometric BRDF kernel exceeds 3% are mostly over lakes, rivers, and coastlines rather than open water. Water surfaces are
flagged using the Terra MODIS Land-Water Mask MOD44W product.

A temperature correction is applied within the air mass factor calculation (e.g., Bucsela et al., 2013) as the slant
column retrievals only use an $NO_2$ absorption cross section at one temperature (294K). The temperature correction factor is
the same factor used in the TROPOMI $NO_2$ product (van Geffen et al., 2019).

Clouds or aerosols are not accounted for in the AMF calculation in this analysis, though cloudy scenes are excluded
from the analysis using a defined count rate threshold measured by the airborne spectrometer detector and visual verification
from GOES 16 imagery (https://www.star.nesdis.noaa.gov/smcd/spb/aq/AerosolWatch/; last accessed 18 April 2020).

Previous work quantified uncertainty in airborne TrVCs from GCAS and GeoTASO by applying error propagation
through the calculation of the vertical column based on uncertainties in the slant column fit, reference spectrum, and AMF
calculation (Nowlan et al., 2016 & 2018; Judd et al., 2019). Relative uncertainties are largest for relatively clean sites (up to
and over 100% in cases), however they decrease as pollution increases. Lorente et al. (2017) found that different methodologies
applied to the same datasets can lead to structural uncertainty of 31-42%, which is mostly due to sensitivity to selection of a
priori vertical profile shapes in the AMF calculation. In this work, airborne TrVCs are evaluated by comparing to Pandora
$NO_2$ columns (Sect. 3) as Pandora $NO_2$ columns have relatively low uncertainties and their AMFs are not dependent on a priori
profile shapes as described in the following section.

**2.3 Pandora spectrometers**

The Pandora instrument is a ground-based UV-VIS spectrometer that provides high-quality spectrally resolved direct sun/lunar
or sky scan radiance measurements. The Pandora radiance measurements combine trace gas spectral fitting routines and, in
the case of sky scan measurements, radiative transfer models to provide column densities of trace gas species similar to
TROPOMI and airborne spectrometers. Pandora measurements obtained throughout the LISTOS study were limited to direct-
sun mode, during which instrument tracks the sun to observe the direct-solar irradiance. Direct-sun columns are particularly
beneficial for validation/evaluation due to their low uncertainties in the AMF (Herman et al., 2009). All data are processed as
part of the Pandonia Global Network (PGN; www.pandonia-global-network.org) and only data with a quality flag of 0 or 10
(high quality) are used. Accuracy and precision of the total $NO_2$ column measurements from Pandora are reported as $2.69\times10^{15}$
molecules $cm^{-2}$/AMF and $1.35x10^{14}$ molecules $cm^{-2}$, respectively (Herman et al. 2009;LuftBlick, 2016). All Pandora data are
converted from total vertical columns to TrVCs by subtracting either the airborne or TROPOMI retrieved stratospheric
columns for comparison purposes.

Ten Pandora spectrometers were deployed and operated in the LISTOS domain in support of the LISTOS air quality study
and as long-term measurements in support of EPA's Photochemical Assessment Monitoring Station Enhanced Monitoring
(PAMS-EM) program (https://www3.epa.gov/ttnamti1/files/ambient/pams/PAMS%20EMP%20Guidance.pdf; last accessed
24 March 2020). Here, we use available Pandora data from nine of the ten instruments between June 2018 and March 2019.
Preliminary analysis indicated that data from one site (City College of New York (CCNY)) had a persistent though variable





low bias relative to airborne data (not shown). This instrument is located on a building rooftop 113 m above ground level and is likely to be missing a portion of the TrVC associated with near surface $NO_2$ that would be observed by downward viewing instruments like TROPOMI and the airborne spectrometers. If co-located with coincident measurements nearer to the surface

(e.g., Nowlan et al., 2016), this missing column could be estimated and applied, but due to the lack of such measurements this site was excluded from analysis. The names, locations, and monthly days of operation of the 9 Pandora spectrometer sites used in this analysis are shown in Table 4. The grey shaded months indicate the time period of the LISTOS study. Figure 1 also shows the spatial distribution of these sites, which includes one site to the west of NYC (RutgersNJ), 3 instruments within the New York City metro area (BayonneNJ, BronxNY, and QueensNY), and 5 along the shoreline of Long Island Sound to the

east-northeast of the city. Pandora sites were chosen to both capture upwind, in-city, and downwind emissions from NYC, particularly $NO_2$ transport down Long Island Sound from the city to help investigate the complex ozone pollution near this land/water interface. All instruments operated during the summer 2018 LISTOS campaign (defined as through September 2018), though four sites operated beyond LISTOS and are used in Sect. 5.2 for evaluation through 19 March 2019.

**2.4 Methods**

All linear regression statistics in this work are calculated using a Reduced Major Axis (RMA) including the coefficient of determination ($r^2$). This regression was chosen over Ordinary Least Squares (OLS) to recognize the potential for uncertainty in both evaluated and reference measurements. Percent and mean differences are also calculated and analyzed and are calculated by the following convention:

Column Difference = evaluated measurement − reference measurement , (1)

Percent (%)Difference = $\frac{\text{Column Difference}}{\text{reference measurement}} \times 100$ , (2)

In Sects. 3 and 5, the reference measurements are the Pandora TrVCs and the evaluated measurements are the airborne and TROPOMI TrVCs, respectively. In Sect. 4, the reference measurements are the aircraft TrVCs and the evaluated measurements are TROPOMI $NO_2$ columns.

For all comparisons, coincidence criteria are chosen based on spatial, temporal, and physical components of the evaluated

and reference measurements. In the following analysis, we use the following coincidence criteria (unless otherwise noted).

- For Pandora and airborne coincidences, the recommended coincidence criteria are from Judd et al. (2019), which are the median airborne TrVCs within a 750 m radius of the Pandora site and the temporally closest Pandora measurement (within ± 5 minutes of the aircraft overpass).
- For airborne comparisons to TROPOMI, each TROPOMI pixel must be at least 75% mapped by cloud-free airborne
pixels within ± 30 minutes of the S5P overpass.
- For Pandora comparisons to TROPOMI, the median Pandora TrVC is calculated within ± 30 minutes of the S5P overpass.
- All TROPOMI data have cloud radiative fractions (CRFs) less than 50%. An additional new criterion is invoked to exclude points for which the difference between surface pressure and cloud pressure in the retrieval (as an indication
of cloud height) exceeds 50 hPa. Justification of this criterion is discussed primarily in Sect. 4.1 and Sect. S3 and the influence of the criterion is considered throughout the paper.



Sensitivities to coincidence criteria are detailed in Tables S1-S3 and briefly discussed in each section and within the supplement to this manuscript.

In addition to the standard TROPOMI v1.2 $NO_2$ TrVC product we consider the effect of using a higher spatial
resolution a priori $NO_2$ vertical profile shape in the TROPOMI retrieval. This is done by recalculating TROPOMI tropospheric AMF using the tropospheric averaging kernel to replace the TM5-MP a priori profile with the 12 km NAMCMAQ data used in the airborne spectrometer AMF calculations following the guidance provided in Sect 8.8 of Eskes et al. (2019).

## 3 Evaluating Airborne TrVC with Pandora Data

This work begins by comparing airborne and Pandora TrVC to evaluate the uncertainty of the airborne TrVCs and establish
the spatial representativeness of the Pandora observations. This evaluation provides a consistent basis for using the high spatial resolution airborne data and high temporal resolution Pandora data to independently assess TROPOMI TrVCs.

During LISTOS, overflights of Pandora sites with the airborne spectrometers occurred during all 13 flight days spanning 25 June – 6 September 2018, between 12:00-22:00 UTC (08:00-18:00 EDT). Site-by-site scatter plots of all coincident measurements and linear regression statistics are shown in Fig. 2. At most sites the Pandora and airborne
tropospheric $NO_2$ columns are highly correlated with slopes of approximately 1. Bars extending from each coincidence illustrate the spatial and temporal variability at the time of the measurements; the horizontal bars show the maximum and minimum Pandora observations within ± 5 minutes of the aircraft overpass and the vertical bars show the 10th-90th percentiles of the airborne pixels within a 750 m radius of the Pandora site (usually ~ 25-30 pixels). High temporal and spatial variations are mostly observed at polluted locations (e.g., QueensNY, BronxNY, and BayonneNJ). NewHavenCT has the lowest slope
(0.71) of all sites yet a high correlation ($r^2$=0.87) which suggests a possible systematic site bias. Such a bias could be due to the inability of the MODIS BRDF product to resolve the spatial gradient of surface reflectance near this site, as this site is adjacent to both a bright urban area in New Haven and also the darker surface of the nearby river. Excluding MadisonCT, which has a poor linear regression due to the few (4) coincidences and small data range, the y-intercepts of the linear regressions range from -1.2 ×10$^{15}$ to 2.0 ×10$^{15}$ molecules cm$^{-2}$. The most likely cause for the range in y-intercepts between sites would be
uncertainty in the estimated column for the reference spectrum in the Pandora retrieval, which uses the Minimum Langley Extrapolation (MLE) approach and has an estimated accuracy of 2.69×10$^{15}$ molecules cm$^{-2}$ for an AMF of 1 (Herman et al., 2009). The observed intercepts are all smaller than this estimated uncertainty.

Figure 3 shows the aggregated comparison of airborne and Pandora TrVC coincidences from all sites during LISTOS (n=171). Figure 3(a) shows the scatter plot and linear regression statistics. Each point is colored by the Pandora location, consistent with Fig. 2. Together, these data are highly correlated ($r^2$=0.92) with a slope of 1.03 and small offset of -0.4×10$^{15}$
molecules cm$^{-2}$. Figure 3(a) also includes whiskers showing the spatial and temporal variability associated with each coincident observation similar to Fig. 2. Two different symbols are used as an objective indicator of temporal variability as quantified by Pandora observations; the outlined squares in Fig. 3(a) are coincidences where the Pandora TrVCs vary less than 30% within



± 15 minutes from the aircraft overpass (n=97) and the non-outlined circles indicate those exceeding 30% (n=74). (The

temporal window for this assessment is larger than the ± 5 minutes shown in the max/min horizontal whiskers to include more

data points to assess temporal variability.)  Most of the temporally homogeneous points tightly span the 1:1 relationship, with

the 95% falling within ± 25 % or have a difference less than $2.69 \times 10^{15}$ molecules cm$^{-2}$. More of the temporally variable points

expand further from the 1:1 line though still mostly fall within ± 50% or have a difference less than $2.69 \times 10^{15}$ molecules cm$^{-2}$

(98%).  Considering only the temporally homogeneous measurements results in a very similar RMA fit (slope and offset) and

a distinctly improved r$^2$ (0.96 vs. 0.92), but a loss of 43% of the number of data points (compare Table S1 Row H to Row B).

This demonstrates the potential benefit of the high temporal resolution of Pandora observations for evaluating the impact of

heterogeneity in NO$_2$ comparisons.

Previous work has suggested that the azimuth direction of the Pandora (due to its sunward viewing observations) can

impact comparisons to airborne spectrometers in heterogeneously NO$_2$ polluted regions (Nowlan et al., 2018; Judd et al., 2019).

We assessed this directionality sensitivity by also examining subsets of the airborne data within sectors surrounding Pandora's

azimuth pointing direction (±22.5 and ±45-degree sectors were considered). The sector constraint slightly degrades the linear

regression statistics, with an increase in slope 4-5%, decrease in y-intercept of $2-3 \times 10^{14}$ molecules cm$^{-2}$, and no change in

correlation (Table S1, compare Rows D and E to Row B). Considering directionality of Pandora can still be important in

assessing individual cases but is not broadly implemented in this analysis due to the relative insensitivity found here and the

limited feasibility of doing it in comparisons with the more spatially coarse measurements from satellites (including

TROPOMI).

While most of the temporally homogeneous points are within ± 25% of each other, there are a small number of

coincidences where the airborne spectrometer retrievals are more than 25% larger than Pandora.  There were no clouds during

these coincidences. The two Bronx coincidences that fall near the 1.25:1 line both occurred on 2 July 2018 during the morning

and afternoon flights. The viewing direction of Pandora toward the southeast in the morning along with elevated NO$_2$ to the

west of the site can partially explain the differences in the morning flight (as indicated by the large vertical whiskers for the

green box near an airborne TrVC of $23 \times 10^{15}$ molecules cm$^{-2}$), though in the afternoon, NO$_2$ is more homogeneous spatially

near this location.  Aerosols are elevated over the site on this day (HALO measured AOT at 532 nm is ~ 0.3), which could

lead to a high bias in airborne TrVCs due to an underestimation in the AMF. However other coincidences during LISTOS also

occurred with AOT of 0.3 or larger and there is no apparent correlation between AOT and the airborne/Pandora differences

(Fig. S1). Other coincidences on July 2$^{nd}$ (n=7) do not show a systematic aircraft high bias.  The other temporally homogeneous

high outlier occurred at Flax Pond on 29 August 2019 just after 13:00 UTC with no explanation related to the viewing direction

of Pandora nor elevated aerosols (AOT ~ 0.16). This coincidence has the lowest calculated airborne tropospheric AMF (0.53),

which may be too low due to the a priori profile being strongly weighted toward the surface than is in reality.  The NAMCMAQ

TrVC at this time is $1.7 \times 10^{15}$ molecules cm$^{-2}$ where 84% of that NO$_2$ is below 300m agl, suggesting too much near-surface

NO$_2$ in this a priori profile.  Less NO$_2$ near the surface in this a priori profile would increase the tropospheric AMF calculation





at this site and a tropospheric AMF of 0.83 would bring this point into agreement with Pandora. The most likely reason for all these differences is incorrect vertical distribution and magnitude of $NO_2$ by the NAMCMAQ model and its influence on the tropospheric AMF (which would need to increase 27-64% to bring these cases into agreement with Pandora).

Figure 3(b) shows the difference between the airborne and Pandora observations as a function of time of day. Overall, there does not appear to be a dependence on time of day, which gives confidence that the airborne retrievals are correctly representing the effects of viewing and solar geometrical input, varying $NO_2$ a priori profiles through the day due to dynamic mixing and the growth of the boundary layer, and varying surface reflectivity based on the MODIS BRDF data in the radiative transfer model. Most (81%) of these differences are within $\pm 2.69\times10^{15}$ molecules cm$^{-2}$—the quoted accuracy of Pandora $NO_2$

retrievals in Herman et al. (2009). These results are encouraging for future validation studies of retrievals from data collected aboard geostationary platforms (e.g., TEMPO; Zoogman et al., 2017) with these types of airborne measurements. Considering only those coincidences during the overpass window of S5P (Table S1, compare Row I to Row B) slightly improves the correlation (r$^2$ increases from 0.93 to 0.94) but degrades the slope and intercept (slope increases from 1.03 to 1.13 with a compensating decrease in the y-intercept from -0.4 to -1.1$\times10^{15}$ molecules cm-$^2$). However, the median percent difference

from Pandora is only 2% during this time period.

        Figure 4 assesses the uncertainty of the airborne data and its potential sensitivity to pollution level. For the least polluted columns (below $3\times10^{15}$ molecules cm$^{-2}$), the interquartile range of the column difference is within $\pm1\times10^{15}$ with a median of $0.1\times10^{15}$. For the more polluted columns, the interquartile range of the percent difference is mostly within 25% with a median difference within $0.6\times10^{15}$ molecules cm$^{-2}$. These conclusions are not dependent on choice of 'reference' (i.e., the

results are similar if examined as a function of binned airborne TrVC). For all data, the median percent difference is -1% with an interquartile range of -23 to 16%.

        Considering all results between Pandora and the airborne spectrometers, uncertainty in the airborne spectrometer TrVC $NO_2$ is generally within $\pm$ 25% with no obvious bias overall. This uncertainty is lower than estimated using error propagation in previous literature, suggesting the errors in a priori datasets are smaller than was estimated in each study

(Nowlan et al., 2016 & 2018; Judd et al., 2019).

## 4 Evaluating TROPOMI TrVC with Airborne Data

Airborne spectrometer data provide a spatially representative dataset in which to compare to TROPOMI with added information about subpixel variability. This is the first airborne spectrometer dataset to be used to evaluate the TROPOMI tropospheric $NO_2$ product.

During the LISTOS campaign, flight plans were designed with the intent to be airborne at the time of the S5P overpass. Figure 5 illustrates how the airborne data are matched to TROPOMI coincidences during three separate orbits—30 June, 19 July, and 6 September. The maps on the top row are true color imagery from the VIIRS sensor which overpasses approximately 5 minutes before S5P (data source: https://worldview.earthdata.nasa.gov/), showing that the first two days were clear of clouds





but cumulus clouds were present during the 6 September overpass. The second row shows the overlaid TROPOMI TrVCs.

$NO_2$ data are colored on a log10 scale spanning 1-100×$10^{15}$ molecules cm$^{-2}$. These three cases illustrate how the day-to-day changes in spatial patterns and the dynamic range of $NO_2$ can be dramatically different from the annual average shown in Fig. 1 (note difference in color bar ranges between Fig. 5 and Fig. 1).

To compare the two datasets, coincident data following appropriate spatial, temporal, and other physical characteristics are extracted as discussed in Sect 2.4. The third row in Fig. 5 shows the airborne data that match the temporal

coincidence criteria for these three orbits (± 30 min from the S5P overpass). The black outlines show TROPOMI pixels that are at least 75% mapped by the airborne spectrometers during this temporal window. Visually, the spatial patterns in TrVC observed by TROPOMI and the airborne instrument are consistent with each other. Finally, the subpixel airborne data within each TROPOMI pixel are gridded to a 250 m matrix to account for overlapping data from adjacent swaths and then the area weighted averages of the airborne TrVCs are computed to create values that are spatially and temporally consistent with the

TROPOMI TrVC observations (bottom row in Fig. 5; gridding methodology from Kim et al., 2016).

From 25 June – 6 September 2018, the airborne spectrometers collected data that coincided with over 1300 TROPOMI pixels within ± 30 minutes of the S5P overpass. However, when considering only pixels 75% mapped by the airborne spectrometer and with CRF less than 50%, the number of coincidences decreases to 621. Additionally, through this analysis, we found that several notable outliers (coincidences with large apparent differences between the two measurements)

corresponded with cloud retrieval effects in cloud-free scenes. Therefore, one additional coincidence criterion is applied to include only scenes with differences between the cloud pressure and surface pressures ($\Delta_{CS}$) less than 50 hPa (the reported uncertainty of the cloud pressure retrieval in van Geffen et al., 2019). This criterion eliminates any TROPOMI pixels with assumed clouds and results in a reduction in the number of data points to 388. The impact of this criterion is discussed in Sect. 4.1 with an illustrative case study in Sect. S3 in the supplemental material, though points exceeding this coincidence criteria

are still shown in scatter plots throughout this paper as blue crosses. (Statistics without this criterion are shown within Tables 5 and 7 and in the supplement).

Figure 6 shows scatterplot and linear regression statistics of all slant and vertical column coincidences between TROPOMI and the airborne data. The red circles in these plots represent the data that meet the strictest coincidence criteria discussed in the previous paragraph. For these points, the slant columns are very highly correlated ($r^2$=0.96), even for cases

with large sub-pixel variation as indicated by the horizontal whiskers in the plot. TROPOMI slant columns are consistently smaller than the airborne spectrometer slant columns (slope=0.59), though airborne slant columns are expected to be larger in comparison to satellite observations because the airborne spectrometers are more sensitive to altitudes nearer to the surface (where much of the $NO_2$ resides) due to the lower observational altitude of the aircraft. However, as shown by the high correlation, TROPOMI and the aircraft are sampling nearly the same atmosphere, at least in the lowest parts of the atmosphere

that make up the majority of the TrVC. Converting from slant to vertical column increases (improves) the regression slope by 15% while preserving the very high correlation ($r^2$=0.96).



While the remaining low bias reflected by the slope below the 1:1 line will be discussed in subsequent sub-sections, we first begin with some discussion about potential reasoning for the small amount of scatter that exists between the TROPOMI and airborne measurements. These causes include: (1) a spatial component (i.e., we allow TROPOMI-scale airborne pixels to

be missing data in up to 25% of the area of the TROPOMI pixel), (2) a temporal component as we allow up to 30 minutes difference between the time of the measurements, and (3) differing a priori assumptions made within each retrieval.

Considering the spatial component of scatter, the horizontal bars in Fig. 6 show the standard deviation of the subpixel airborne TrVCs within each TROPOMI pixel. Generally, the variation in subpixel $NO_2$ increases as the $NO_2$ TrVC increases, illustrating how scatter in the comparisons could increase if only small subsets of the pixel are mapped. Sensitivity to the

mapped percentage is annotated in Table S2 (rows B-D and M-O) and shows little impact when relaxing the percent-mapped criterion to 50% (though is impacted negatively when the $\Delta_{CS}$ criterion is applied (Table S2: rows M-O)) and a more significant decrease when relaxing to 25%. At least with the airborne samples in this case, the linear statistics are driven by the most polluted pixels that are 100% mapped by the airborne spectrometers, explaining the limited sensitivity in the RMA fit to the percentage of the TROPOMI pixel mapped in this study.

Addressing the temporal component, if the temporal window is decreased to ± 15 minutes from ± 30 minutes, the number of mapped TROPOMI pixels by the aircraft decreases by 65% while the quality of linear statistics is moderately improved (Table S2, compare Row B to Row E). However, there is a larger adverse impact to the RMA fit and $r^2$ when the time window is extended to extract airborne data within ± 60 minutes of the S5P overpass. Coincidences occurring between 30-60 minutes of the S5P overpass are shown as open circles in Fig. 6. For example, the small subset of very polluted airborne

TrVCs that are much larger than what is retrieved by TROPOMI occurred during a time with high temporal variability on 2 July 2018. The airborne spectrometer observed a distinct very polluted plume over NYC and over the 48-minute period between the airborne and TROPOMI observations, a Pandora spectrometer located at the CCNY observed a 50% decrease in $NO_2$ total vertical column, leading to a large difference between the airborne and TROPOMI TrVCs when the temporal window is extended to ± 60 min. (Note that because the CCNY Pandora is placed well above the surface it was excluded from the

airborne and TROPOMI comparisons and no other Pandora instruments coincided with this feature). These outliers are caused by real spatiotemporal variability rather than issues in either of the retrievals and demonstrate the care needed for matching airborne data collected over time to the nearly instantaneous observations from S5P TROPOMI. These large differences are also apparent in the slant column comparisons and future studies should consider slant column comparison between aircraft and TROPOMI as a guide for identifying potential spatial and temporal mismatches.

With respect to differing retrieval assumptions, we consider two factors in the following subsections: treatment of clouds and $NO_2$ vertical profile shape.



## 4.1 Cloud retrieval effects

In previous literature, a coincidence criterion based on CRF from TROPOMI has been the common consideration for data comparisons, though studies vary slightly in their chosen CRF threshold (ranging from 30-50% in Griffin et al. (2019), Ialongo

et al. (2020), and Zhao et al. (2020)). We investigate the effect on the statistics of varying CRF threshold, alone, but find that retrieved cloud height is also an important factor and here consider the two effects together.

In the TROPOMI retrieval, surface reflectivity is estimated using the $0.5° \times 0.5°$ climatology from five years of OMI observations (Kleipool et al., 2008; van Geffen et al., 2019). When the surface albedo climatology used for TROPOMI has a low bias, which can occur over bright city centers, the algorithm increases the overall brightness of the scene by assuming a

non-zero cloud fraction. In cloud-free urban scenes, this approach generally results in a non-zero CRF with a nominal cloud pressure equal to the surface pressure. Fig. S2(a) illustrates this behavior on a cloud-free day (19 July 2018).

This CRF-adjustment approach over bright surfaces generally appears to work well, however we identified a potential issue when the retrieval also places retrieved "clouds" above the surface rather than at the surface in cloud-free scenes. The two most obvious illustrations of this effect are evident as the two blue crosses farthest above the regression line with airborne

TrVCs greater than $25 \times 10^{15}$ molecules $cm^{-2}$ in Fig. 6. Section S3 in the supplemental material presents a case study demonstrating that the effect is correctable for these two points. We note that in the presence of significant scattering aerosols, CRF may also be larger than zero and the cloud pressure level may mimic the height of the aerosol layer, however in this case elevated aerosol has been ruled out by the HALO measurements co-located with the airborne spectrometer. Clouds and their effect on the estimated vertical sensitivity are an important component within the $NO_2$ retrieval, as clouds are assumed to

'shield' the view of the atmosphere below the cloud level in some fractions of the pixel. However, in cloud-free scenes, cloud pressures significantly less than the surface pressure with elevated CRF can lead to an underestimation in the AMF, and therefore an overestimation in TROPOMI TrVC, as the shielding that is assumed through the retrieval is not occurring in reality. Because the airborne screening criteria ensure that only cloud-free observations are included in our analysis, our comparisons are biased toward cloud-free scenes, and therefore high CRFs are associated generally with bright surfaces instead

of clouds.

To avoid these impacts, we explored an additional coincidence criterion based on cloud parameters in the TROPOMI product file. We consider an allowable difference between retrieved cloud pressure and surface pressure (henceforth $\Delta_{CS}$) of less than 50 hPa (which is the reported uncertainty in cloud pressure retrieval from van Geffen et al., 2019). Figure 6 shows points that exceed this criterion as blue cross symbols and the linear regression statistics with and without this criterion applied

are summarized in Table 5. Applying this criterion removes approximately 30% of coincidences including the largest outliers but also many points that are not outliers. Of the 233 data points that have $\Delta_{CS}$ greater than 50 hPa, 58% (n=136) of them have aircraft measured cloud fractions of less than 2%, and 69% of these cloud free coincidences (n=94) have reported CRFs greater than 10%, illustrating that the cloud retrieval regularly yields an effective cloud height above the surface even during cloud-free scenes. Further filtering data by only removing data with CRFs > 10% results in very little change in the overall statistics.



Table 5 shows that the largest impact of the $\Delta_{CS}$ criterion is an improvement in the correlation ($r^2$ of 0.96 vs. 0.90) but a slope further from 1 (0.68 vs. 0.71) and a more negative median percent difference (-19% vs -11%), showing that there is excellent correlation between the two measurements but an apparent low bias in the TROPOMI retrieval that the cloud pressure errors partially offset. This impact is also confined to the TrVC comparisons and not apparent in the slant column comparisons, which demonstrates the impact is through assumptions made in the AMF calculation.

495        Eskes and Eichmann (2019) mention occurrences of negative effective cloud fractions in the FRESCO cloud product that could also result in positive cloud fraction in the $NO_2$ window in v1.2 of the TROPOMI TrVC product which causes a noisy $NO_2$ retrieval. The occurrence of negative FRESCO cloud fractions with positive CRFs did occur during many of these coincidences (63% of the 621 pixels). However, this fraction is much lower for $\Delta_{CS}$ flagged pixels (18%) and they were not associated with the largest outliers in this analysis. Applying a criterion to remove negative cloud fractions instead of $\Delta_{CS}$

flagged pixels results in similar results as only filtering for CRFs < 50% and no $\Delta_{CS}$ criterion (slope=0.72, offset=$0.7\times10^{15}$ molecules cm$^{-2}$, $r^2$=0.91, and n=233). Therefore, this impact is not the cause for the described patterns in the previous paragraph.

        In the vertical columns, coincidences identified by the $\Delta_{CS}$ criterion typically lie above the best-fit line, consistent with the hypothesis of effective cloud shielding in the AMF calculation during cloud-free scenes. There is one obvious

coincidence exceeding the $\Delta_{CS}$ threshold that opposes this general pattern by falling below the best fit line (blue cross with airborne TrVC around $50\times10^{15}$ molecules cm$^{-2}$). This apparent disparity appears to be caused by large temporal variation between the times of the airborne and satellite measurements. The airborne measurement preceded TROPOMI by 23 minutes and in a subsequent airborne measurement over the same area 70 minutes later, the airborne $NO_2$ TrVC had decreased to approximately $30\times10^{15}$ molecules cm$^{-2}$, which is much nearer to the TROPOMI-measured value of $25\times10^{15}$ molecules cm$^{-2}$.

This is another example where a temporal mismatch resulted in an outlier in the slant column comparisons in Fig. 6(a) demonstrating the use of slant column comparisons to assist in identifying spatial and temporal mismatches.

        Finally, we summarize the sensitivity to different CRF thresholds. Without the $\Delta_{CS}$ criterion applied (Table S2; Rows F-I), allowing larger CRF values generally decreases $r^2$ while increasing the slope slightly and dramatically increasing the number of coincidences. Highest correlations, up to 0.96, are maintained with CRF < 20%. When the $\Delta_{CS}$ threshold is applied, the

RMA fit is largely insensitive to changes in CRF up to 50% (Table S2: Rows J-M), maintaining the high quality of the linear regression while including progressively more data points with increasing CRF thresholds. Because CRF can often exceed 20% over urban areas even in cloud free conditions due to effects of the coarse a priori surface reflectivity used in the retrieval, the $\Delta_{CS}$ criterion appears useful for retaining valid cloud-free coincidences over bright urban scenes. Overall, the best fit is attained either by restricting CRF to less than 20% and not using the $\Delta_{CS}$ criterion or by using the $\Delta_{CS}$ criterion, which allows

inclusion of CRF values up to 50% and provides 35% more coincidences. Future research could explore using alternative cloud measurements (e.g., from VIIRS) to identify cloud-free scenes and the use of clear-sky AMFs.



## 4.2 NO₂ vertical profile shape

The a priori vertical profiles in the TROPOMI NO₂ retrieval are from the TM5-MP model with a spatial resolution of 1° × 1° interpolated to the center of the TROPOMI pixels (van Geffen et al., 2019). In a heterogeneously polluted region such as

NYC, NO₂ profiles vary at much smaller spatial scales. For spatial reference, the area flown by the airborne spectrometer flights for each LISTOS raster (Fig. 1) cover an area of approximately 1° × 1° or smaller and airborne TrVCs span up to two orders of magnitude in this domain. Here, TROPOMI tropospheric AMFs are recalculated with the 12 km NAMCMAQ analysis used in the airborne TrVC retrieval to demonstrate the impact of spatial resolution of a priori profiles. These TROPOMI TrVCs columns are hereafter labeled as TROPOMI-NAMCMAQ. The original TROPOMI v1.2 product is referred

to as TROPOMI Standard.

Figure 7 has the same format as Fig. 6 but instead compares TROPOMI-NAMCMAQ to airborne TrVCs. (Note that both datasets are now using the same a priori profiles.) In general, applying the NAMCMAQ profile into the TROPOMI AMF calculation brings the airborne and TROPOMI data into closer agreement; with the $\Delta_{CS}$ criterion applied, slope increases 13% from 0.68 to 0.77, the median percent difference improves from -19% to -7%, and a high $r^2$ is maintained (changing from 0.96

to 0.95).

Incorporating a higher resolution a priori profile appears to result in an increase in the sensitivity to the $\Delta_{CS}$ criterion, with more of the blue cross points visible in Fig. 7 than in Fig. 6, which can likely be attributed to increased sensitivity to the lower altitude levels in the AMF calculation. In the higher resolution NAMCMAQ analysis, the lower levels are more polluted and thus more sensitive to 'cloud' shielding.

The biases of the TROPOMI Standard and TROPOMI-NAMCMAQ TrVCs with respect to the airborne data are further examined as a function of pollution level in Fig. 8. The majority of points (68%) are less than $6\times10^{15}$ molecules cm⁻², so the overall distributions are dominated by the behavior in the lowest bins in Fig. 8. In these lowest two bins, the median percent difference is -10% and +3%, respectively for TROPOMI Standard and TROPOMI-NAMCMAQ TrVCs. Column differences unsurprisingly increase with pollution level and are small in these two lowest bins, with the interquartile range

within $1\times10^{15}$ molecules cm⁻² and inner 90% of points having differences within $2\times10^{15}$ molecules cm⁻². TROPOMI Standard has a median absolute bias of zero in the lowest bin. Using the NAMCMAQ profile shifts the bias more positive in all bins, creating a small positive bias in the lowest bin but reducing the overall median bias from $-1\times10^{15}$ molecules cm⁻² to $0.3\times10^{15}$ molecules cm⁻². For airborne TrVCs above $6\times10^{15}$ molecules cm⁻², the median percent difference is -29% for the TROPOMI Standard but improves to -20% for TROPOMI-NAMCMAQ. Although a higher resolution a priori profile improves the overall

bias in the TROPOMI product, there is still a low bias for the most polluted TROPOMI TrVCs columns.



## 5 Evaluating TROPOMI TrVC with Pandora Data

Pandoras operated in the LISTOS domain during and after the conclusion of the intensive LISTOS airborne measurements as part of the PAMS EM Program (see Table 4). Following coincidence criteria in line with those from Sect. 4 (TROPOMI CRF < 50%, $\Delta_{CS}$ less than 50hPa, and median Pandora TrVC within ±30 minutes), Fig. 9 shows all coincidences between Pandora and TROPOMI through 19 March 2019, with coincidences during the LISTOS intensive period (defined as any measurements prior to and including 30 September 2018) outlined in black. Site-by-site statistics are listed in Table 6 for both time periods. In this section we discuss consistency in TROPOMI evaluation results with airborne spectrometers using data from only the LISTOS time period and also from an extended temporal window at select sites that operated through winter 2019.

### 5.1 TROPOMI v. Pandora during LISTOS

During the LISTOS time period, there were 156 coincidences between the nine Pandora spectrometers and TROPOMI, ranging from 8 to 25 coincidences by site (Table 6). With the exception of MadisonCT and BranfordCT (which lack in TrVC dynamic range), the slope of TROPOMI vs. Pandora is less than one (ranging from 0.49-0.84, similar to the results in Sect. 4) with moderate to high values of $r^2$ (0.29-0.90). All median percent differences are negative and vary by site ranging from -9% to -52%.

Figure 10(a) shows the aggregated TROPOMI Standard and Pandora dataset during LISTOS; red circles/blue crosses are those that have a $\Delta_{CS}$ less than/greater than 50hPa, respectively, similar to Fig. 6. The bars represent the reported precision of the TROPOMI Standard product (vertical) and the $10^{th}$-$90^{th}$ percentile of Pandora data within the ± 30 min window (horizontal). The aggregated dataset shows that TROPOMI TrVCs have a low bias in comparison to Pandora (slope=0.80 and offset of $-0.7 \times 10^{15}$ molecules cm$^{-2}$) and high correlation ($r^2$=0.84). As a whole, TROPOMI has a median percent difference from Pandora of -33% with an interquartile range of -48% to -14%, consistent with comparisons of TROPOMI to airborne TrVCs for values above $6 \times 10^{15}$ molecules cm$^{-2}$. Comparing Fig. 10(a) to Fig. 6(b), the slope is 18% higher (better) than in the comparisons to the TROPOMI Standard product to airborne TrVCs, though at the expense of a lower $r^2$ (0.96 vs. 0.84). Coincidences at QueensNY and BronxNY have the lowest median percent difference of all the sites and the aggregate slope is sensitive to whether these two sites are included or not (0.80 and 0.72 with and without BronxNY and QueensNY, respectively). This result highlights the sensitivity of site selection and duration in the combined analysis and can likely be attributed to differences in spatial representivity between the TROPOMI and Pandora and perhaps sampling temporally over just the short period of the LISTOS study.

Spatial representivity of Pandora and sub-pixel variation in the TROPOMI area can also influence the results. TROPOMI pixels span an areal coverage of approximately 30-130 km$^2$ depending on the position in the swath through S5P's 16-day orbit cycle, while Pandora measurements represent a more localized environment. We found that the interquartile range of the TROPOMI bias relative to Pandora becomes slightly more negative as the pixel size gets larger (not shown). For pixels less than 40km$^2$, the interquartile range is -1% to -46% (n=67), whereas for pixels larger than 80km$^2$, it is -14% to -59% (n=18).





Although cloud information for Pandora comparisons at TROPOMI sub-pixel resolution is not readily available, the impact of coincidence criteria based on clouds is assessed similarly to Sect. 4. Lowering of the CRF threshold preferentially
excludes data from sites with brighter surface reflectivity and, typically, larger $NO_2$ values. For example, QueensNY has a median CRF of 34% (minimum of 17%), whereas a more rural location like WestportCT has a median CRF of 8% (minimum of 0%). Without applying the $\Delta_{CS}$ criterion, we find the quality of the linear regression statistics to be quite sensitive to CRF threshold (Table S3, Rows F-I). Using more restrictive CRF thresholds generally worsens the correlation and the trends here are less consistent than found in the TROPOMI-airborne comparisons. This inconsistency is due to the relatively fewer number
of Pandora coincidences having large values, e.g. above $10\times10^{15}$ molecules cm$^{-2}$, which makes the linear regression sensitive to screening criteria such as CRF that exclude any of the larger-valued data points. Though applying the $\Delta_{CS}$ criterion removes nearly half the coincidences for CRFs < 50%, its application increases $r^2$ values at all CRF thresholds (Table S3; Rows J-M). Applying the $\Delta_{CS}$ criterion maintains high correlations while allowing retention of data from bright urban sites that would be preferentially left out by filtering by CRF for thresholds 30% and lower.

Figure 10(b) shows the comparison between TROPOMI-NAMCMAQ TrVCs and Pandora. Many more coincidences with $\Delta_{CS}$ greater than 50hPa (blue crosses) are evident above the 1:1 line, again illustrating the increased sensitivity to this parameter when higher resolution a priori profiles are used within the TROPOMI AMF calculation. Table 7 summarizes all the various cases. Considering all coincidences without invoking the $\Delta_{CS}$ criterion (i.e., including blue crosses and red circles), there is a large improvement in the regression statistics from TROPOMI Standard to TROPOMI-NAMCMAQ, with the slope
closer to 1 and a median percent difference of only -9% (relative to the -30% for TROPOMI Standard). However, as illustrated by the blue points in Fig. 10(b), it is clear that this 'improvement' is partially driven by a high bias related to the impact of clouds. When points with $\Delta_{CS}$ greater than 50hPa are excluded, the slope between TROPOMI-NAMCMAQ and Pandora improves by only 2.5% in comparison to TROPOMI Standard with a slight degradation of $r^2$ from 0.84 to 0.80. However, there is a large improvement in the median percent difference, from -33% (interquartile range of -48% to -14%) for TROPOMI
Standard to -19% (interquartile range of -36 to 5%) for TROPOMI-NAMCMAQ.

Much of the correlation in Fig. 10 is driven by the 20 points above $10\times10^{15}$ molecules cm$^{-2}$; considering only points below $10\times10^{15}$ molecules cm$^{-2}$ lowers $r^2$ to 0.42 and 0.39 for TROPOMI Standard and TROPOMI-NAMCMAQ, respectively, though results in the same median percent differences. The loss in correlation demonstrates the challenge of doing linear regressions on datasets with a lack of dynamic range well above $10\times10^{15}$ molecules cm$^{-2}$ in this analysis when spatiotemporal
variability impacts can be at a similar magnitude. However, extending analysis through winter 2019 results in a larger sampled dynamic range as demonstrated in the next section.

## 5.2 TROPOMI v. Pandora through 19 March 2019

The deployment of many of the Pandora instruments in this region as part of the PAMS EM Program presents the opportunity for evaluation beyond the period of the LISTOS intensive campaign. TROPOMI level 2 $NO_2$ processing switched to version





1.3 after 19 March 2019, thus this analysis goes only through this date to avoid possible influences associated with the version change. To ensure consistent spatial representivity through the period, analysis is limited to the four sites that continued operation through 19 March 2019 (Table 4; RutgersNJ, BayonneNJ, QueensNY, and WestportCT).  The focus of this extended analysis is to see whether conclusions made from the LISTOS time period are still valid through the fall and winter months as photochemistry and meteorological changes lead to potential shifts in spatial and temporal variation and dynamic range at

these sites. These four sites represent two in-city sites and sites upwind and downwind from NYC, though the upwind/downwind side of the city is dependent on wind direction from day-to-day. Figure 11 shows timeseries of Pandora and TROPOMI Standard TrVCs from 25 June 2018 through 19 March 2019 at each of the sites.  Colored circles represent the Pandora measurements during the S5P overpass, the black stars show the TROPOMI TrVC, and the whiskers indicate variability or uncertainty (see figure caption). Note that some days have two overpasses. In general, temporal patterns are

similar in both TROPOMI and Pandora measurements demonstrating each instruments ability to observe synoptic and seasonal variability in TrVCs.

At RutgersNJ and WestportCT, Pandora and TROPOMI TrVCs rarely exceed $10\times10^{15}$ molecules cm$^{-2}$ during the year.  More polluted coincidences occurred periodically during November-March as expected given the longer photochemical lifetime of NO$_2$ during winter. In early January, when both Pandora and TROPOMI values were low, the spatial distribution

of NO$_2$ in the LISTOS domain from TROPOMI showed that the NYC plume was advected over the Atlantic Ocean on most of these days and was not intercepted by either site. At WestportCT, there was an extended period of elevated columns near the end of January and beginning of February. The larger TrVC values during that period coincide with days when the NYC plume extends toward Long Island Sound and Connecticut, likely driven by synoptic flow from the southwest quadrant. (This is the flow orientation that is often linked with poor ozone air quality along the shorelines of Long Island Sound during the

summertime, e.g., the late August 2018 timeframe which was active with respect to ozone (airnow.gov: last accessed 11 March 2019) but did not result in an NO$_2$ enhancement over WestportCT, likely due to the shorter NO$_2$ lifetime in summer.) Alternatively, at RutgersNJ on the 9th of March, the Pandora site was encompassed by an NO$_2$ plume extending from the center of NYC during two consecutive TROPOMI overpasses leading to its maximum TrVC values during the time period assessed. Unlike the other two sites, BayonneNJ and QueensNY have large dynamic ranges in NO$_2$ TrVCs in all seasons due to their

proximity to strong sources within the NYC metropolitan area. Extending comparisons through the winter allows for more frequently measuring large values to extend the dynamic range of the coincident measurements.

Figure 11(e) shows the percent difference in TROPOMI TrVCs from Pandora with the bars showing the temporal variability of these percent differences during the ± 30-minute temporal window from the S5P overpass (10th-90th percentile). Despite some changes seasonally in the magnitude of NO$_2$ at each of the sites, the percent difference in TROPOMI from

Pandora does not have an apparent significant trend over this time period. The majority of points fall within 0% to -50%. The points with percent differences closest to zero, including points with positive percent differences, are associated with small values at WestportCT.   Many of the coincidences have very large ranges in percent difference due to the temporal variability





of Pandora TrVCs within the ± 30-minute time period that are likely associated with sub-pixel heterogeneity, again illustrating the challenge of quantifying biases with Pandora in urban environments.

650         Figure 12 shows a scatter plot of the coincidences at these four sites during both the LISTOS timeframe (Fig. 12(a)) and the longer 9-month period (Fig. 12(b)). During the LISTOS period the slope is 0.76 and a reasonably high $r^2$ of 0.89 is caused by the large range of TrVCs observed at BayonneNJ and QueensNY. These results are similar to those at all nine locations during the LISTOS timeframe (Fig. 10(a)) with the same median percent difference. The number of coincidences through the LISTOS months is low (n=58) due to the $\Delta_{CS}$ threshold being frequently exceeded (Table 7). The number and

dynamic range of observations is greater when extended through the rest of the year (n=195). The overall median percent difference is 8% lower over the 9-month period (-27%) than the LISTOS timeframe (-19%), and though it is not visually apparent in Fig. 11(e), this drop is reflected by a decrease in the median percent difference at QueensNY (Table 6). At QueensNY, the median percent difference for TrVCs becomes more negative at higher magnitudes of TrVC; Pandora TrVCs less than/greater than $15\times10^{15}$ molecules cm$^{-2}$ have a median percent difference of -15% and -33%, respectively, at this site.

Despite large day-to-day variations and changes in dynamic range through the seasons, the linear statistics for the aggregated data at these four sites are largely unchanged when comparing the LISTOS time frame to the extended 9-month period (2.5% difference in slope and 0.01 range in $r^2$).

## 6 Overall evaluation of TROPOMI v1.2 NO₂ TrVCs

Tables 5 and 7 summarize the overall results of TROPOMI TrVC comparisons to the airborne and Pandora spectrometers from

this work. No matter the reference dataset or data selection criteria, linear regression and percent difference statistics indicate that in this urban coastal region the v1.2 TROPOMI Standard TrVC product has a low bias. Median TROPOMI NO₂ TrVCs are 19% and 33% lower than airborne and Pandora TrVCs, respectively, during the LISTOS timeframe. These different values are partially related to the characteristics of sampling at different TrVC ranges between the two datasets. One-third (130) of the airborne coincidences have TrVC less than $3\times10^{15}$ molecules cm$^{-2}$ with no observed bias between the two measurements,

while only 19 of the 156 Pandora coincidences have TrVC less than $3\times10^{15}$ molecules cm$^{-2}$ with TROPOMI having a low bias of -21% at these cleanest levels. At higher TrVC magnitudes (greater than $6\times10^{15}$ molecules cm$^{-2}$), the percent differences of TROPOMI from aircraft (-29%) and Pandora (-31%) are more similar to each other. Lesser polluted columns are more sensitive to uncertainties related to the stratospheric columns, references, and other assumptions (which are different between all retrievals), whereas at more polluted levels the bias is more attributed to uncertainties in tropospheric air mass factors.

675         Overall these results are consistent with other studies using Pandora spectrometers to evaluate the TROPOMI NO₂ products, as they also found that the TROPOMI NO₂ product has a low bias in the Canadian Oil Sands (Griffin et al., 2019), Toronto, Canada (Zhao et al., 2019), Paris, France (Lorente et al., 2019), and polluted scenes ( > $10\times10^{15}$ molecules cm$^{-2}$) near Helsinki (Ialongo et al., 2020). Many of these studies found improvement by using higher resolution regional model a priori profile shapes in the AMF calculation for TROPOMI. In this study, recalculating the TROPOMI tropospheric AMF with the





higher resolution 12 km NAMCMAQ analysis resolves some of the low bias in TROPOMI TrVCs, improving median percent differences from -19% to -7% with respect to airborne data and from -33% to -19% with respect to Pandora data. However, despite this improvement, there is still a persistent low bias in the TROPOMI TrVCs.

      This analysis is impacted by influences of cloud pressure in the TROPOMI retrieval. Invoking the $\Delta_{CS}$ criterion increases (worsens) the overall TROPOMI low bias as it removes a high bias caused by assumed cloud shielding in the AMF calculation

in cloud-free scenes. In all comparisons shown in Tables 5 and 7, the median percent difference is more negative (worse) when only points with $\Delta_{CS}$ less than 50 hPa are included, and the effect is more pronounced for TROPOMI-NAMCMAQ coincidences (decreasing 10-11%) than for TROPOMI Standard (decreasing 4-8%). Invoking the criterion also consistently improves the correlation in every case by removing many of the outlier points, as intended. The most striking examples are the airborne comparison with TROPOMI-NAMCMAQ ($r^2$ improved from 0.83 to 0.95) and Pandora comparison with

TROPOMI-Standard for the 4-site subset of the LISTOS period ($r^2$ improved from 0.79 to 0.88).

**7 Conclusions**

The operational nature of the S5P TROPOMI mission as part of the Copernicus Program marks an important step forward in monitoring of the environment, amplifying the need for increased validation capacity of satellite trace gas data. The datasets collected in support of the Long Island Sound Tropospheric Ozone Study during summer 2018 and as part of the PAMS-EM

program are exceptional for evaluation of TROPOMI TrVCs, providing a robust set of independent remotely sensed $NO_2$ column densities from airborne spectrometers (13 mapping flights from 25 June 2018 to 6 September 2018) and a network of nine ground-based Pandora spectrometer systems.

      Previous studies have shown that Pandora direct-sun $NO_2$ columns are valuable for validating airborne spectrometer retrievals due to their high precision and temporal resolution and comparable spatial resolution. In this study, the airborne

spectrometer data are highly correlated with Pandora measurements with a slope of 1.03, offset of $-0.4\times10^{15}$ molecules cm$^{-2}$, and $r^2$=0.92. Much of the remaining scatter in the data can be attributed to the spatiotemporal heterogeneity of $NO_2$ in this urban coastal environment, as evaluating only the less temporally varying measurements shows similar statistics but a higher $r^2$ of 0.96. Though singular comparisons can exceed differences of 25%, overall the majority of the coincidences fall well within ±25% and 81% of the coincidences fall within the reported accuracy of Pandora of $2.69\times10^{15}$ molecules cm$^{-2}$. These

results give confidence for using both datasets to assess the TROPOMI TrVC product.

      The combination of these two reference measurements in one region presents unique strengths for validation of TROPOMI TrVCs over a domain with large variations in $NO_2$. Pandora measurements are useful for evaluating space-based and aircraft-based retrievals due to their ability to observe continuously in one location for long time periods. However, the impact of subpixel heterogeneity within satellite sub-pixel areas can lead to mismatches between the Pandora and satellite

despite the much-improved spatial resolution of TROPOMI. Airborne spectrometers are typically only deployed for short periods of time, but their observations are more spatially representative of the satellite measurements with the added capability



of retrieving at subpixel resolutions over the entire TROPOMI pixel areas they overfly. In this study, strengths of the two reference measurements were able to be combined. TROPOMI comparisons to airborne TrVCs are more correlated than Pandora comparisons during the LISTOS timeframe ($r^2$=0.96 vs. 0.84). Additionally, the long-term deployment of Pandora
instruments as part of the PAMS-EM program allowed TROPOMI TrVCs to be assessed over multiple seasons. We find the strongest impact of seasonality is the extension of the TrVC dynamic range sampled in the winter months, providing more robust statistical fits though not very significant changes in the statistics overall between the two time periods.

During the LISTOS timeframe, TROPOMI Standard TrVC data have a low bias in comparison to Pandora and airborne TrVCs of -33% and -19%, respectively. This bias improves to -19% and -7% when TROPOMI TrVC is recalculated
using an AMF with the 12 km NAMCMAQ a priori profile. These results are obtained by screening out cases where cloud shielding estimated in the TROPOMI retrieval occurred over cloud free scenes, which tend to compensate partially for the TROPOMI TrVC low bias amd introduce significant artifacts that degrade correlations with reference measurements. Future exploration of cloud-based coincidence criteria would help in identifying effects of cloud parameters on $NO_2$ trace gas comparisons and other evaluations of near-surface weighted trace gases such as HCHO. It will also help in evaluating how
these sensitivities change as cloud retrievals and their implementation into the trace-gas retrievals evolve in future versions (e.g., in v1.3, implemented after 19 March 2019, the FRESCO-S cloud retrieval was updated adjust surface albedo in cloud-free areas where the surface albedo climatology is too low, as discussed in Eskes and Eichmann, 2019).

We find the v1.2 TROPOMI Standard TrVCs to be within the validation requirements for the mission (bias within ± 25-50%; van Geffen et al., 2019) but with a persistent low bias in the NYC region. While some of the low bias is removed by
the incorporation of a higher resolution a priori vertical profile, there is still a low bias in the TROPOMI $NO_2$ TrVC retrieval which indicates the need for improved a priori assumptions in the AMF calculations. This analysis looked at the impacts of a priori $NO_2$ profiles at a moderately higher resolution and of clouds, and future work should also explore effects of surface reflectivity. Some differences between TROPOMI and airborne TrVCs can be related to differences in a priori assumptions between the TROPOMI and airborne retrievals; Lorente et al. (2017) discussed that the structural uncertainty in tropospheric
air mass factors is up to 42% in polluted regions due to different retrieval methodologies. Future comparisons should consider using common methodologies for AMF calculation for both airborne and TROPOMI TrVCs to better quantify the sensitivity of specific a priori assumptions in AMF calculations.

This work is the first dataset that has used airborne spectrometer measurements to evaluate a satellite $NO_2$ retrieval at this degree of success (large number of coincidences, high correlation, and mapping 100% of pixel areas) which is supportive
of using airborne spectrometers to validate and evaluate UV-VIS trace gas retrievals during current and future satellite missions, including geostationary measurements. This LISTOS dataset, as well as future ones collected during other intensive field studies, will be useful for continuing to evaluate the TROPOMI algorithms through future releases of the TROPOMI $NO_2$ TrVC product.



**Author Contribution**

LMJ prepared the manuscript with contributions from all coauthors. JAA, LCV, JJS, RBP, and LMJ led flight planning activities for LISTOS. SJJ, MGK, and LMJ collected the airborne spectrometer data and AN collected HALO data during LISTOS flights. LMJ processed the airborne spectrometer NO$_2$ retrievals. RBP provided the NAM-CMAQ analysis used in the vertical column retrieval and in reprocessing of TROPOMI data. CRN and GGA provided the Smithsonian Astrophysical Observatory AMF Tool as well as guidance in its use for AMF calculations. HJE and JPV provided their expertise in the TROPOMI product and discussed results periodically through this project. JJS, DW, LCV, and RS led the coordination, installation, and maintenance of Pandora spectrometers in the LISTOS domain. AC, MM, and MG led the processing of the Pandora NO$_2$ retrievals and provided guidance in Pandora data analysis.

**Acknowledgements**

Authors would like to acknowledge Peter Pantina and Sanxiong Xiong for their participation in airborne data collection during LISTOS flights, members of the HALO team for supporting flights and data processing of aerosol optical depth, Nader Abuhassan and Lena Shalaby for their assistance in installing and monitoring the Pandora network in the LISTOS domain, extending to the larger Pandora teams at NASA GSFC and LuftBlick through their support in Pandora data processing. The LISTOS airborne measurements would not have been possible without the support of the NASA GEO-CAPE Mission Study as well as NASA ESD Tropospheric Composition Program. We express gratitude to the entire LISTOS science team for their expertise, research, and measurement contributions toward the successful collaborative field study.

This work is done in part through the Sentinel-5P Validation Team Projects 28695 and 40030. This work contains modified Copernicus data.

Disclaimer: The research described in this article has been reviewed by the U.S. Environmental Protection Agency (EPA) and approved for publication. Approval does not signify that the contents necessarily reflect the views and the policies of the agency nor does mention of trade names or commercial products constitute endorsement or recommendation for use.

**Data Availability**

TROPOMI: https://s5phub.copernicus.eu/dhus/#/home

Airborne spectrometer data version R0: https://www-air.larc.nasa.gov/cgi-bin/ArcView/listos

Pandora data can be found at www.pandonia-global-network.org. QueensNY, BayonneNJ, and BronxNY have been processed with versions rnvs1p1-7 and the rest of the sites were processed with rnvs0p1-5. On the official PGN webpage just the nvs1p1-7 data will be accessible as soon as the data is available. There is no difference between the products except the data flagging procedures.



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



**Table 1: A priori input for tropospheric AMF calculations for TROPOMI and airborne TrVCs**

|  | **TROPOMI v1.2** | **Airborne** |
|---|---|---|
| **A priori NO₂ profile shape** | TM5-MP 1° × 1° model (Williams et al., 2017) | Troposphere: 12 km NAMCMAQ (Stajner et al., 2011) Stratosphere: PRATMO Climatology (Prather, 1992; McLinden et al., 2000) bias corrected daily with TROPOMI Stratospheric Vertical Columns |
| **Surface Reflectivity** | OMI 0.5°× 0.5° 5-year climatology (Kleipool et al., 2008) | Land: MCD43A1 daily L3 500m v006 product (Lucht et al., 2000; Schaaf and Wang, 2015) averaged over the period of the campaign Water: Assumed Lambertian reflectance of at least 3% and Cox-Monk kernel |
| **Pressure/Temperature Profiles** | TM5-MP 1° × 1° model driven by the ECMWF corrected with a 3-km DEM | Troposphere: 12 km NAMCMAQ (Stajner et al., 2011) Stratosphere: 1° RAQMS (Pierce et al., 2009) |
| **Clouds** | FRESCO-S (Loyola et al., 2018) | Cloudy scenes are not included in this analysis |


**Table 2: Comparison of GeoTASO and GCAS**

|  | **GeoTASO** | **GCAS** |
|---|---|---|
| **Spectral Range** | 290-390nm, 415-695nm | 300-490nm, 480-900nm |
| **Spectral Resolution** | 0.43nm, 0.88nm | 0.6nm, 2.8nm |
| **Size/Weight** | 200 lbs | 80 lbs |
| **Detector dimensions** | 1056 spectral × 1033 spatial | 1072 spectral × 1024 spatial |
| **Native spatial resolution** | Approximately 250 m × 250 m | |
| **Field of View** | 45 degrees | |
| **References** | Leitch et al., 2014 Nowlan et al., 2016 Judd et al., 2019 | Kowalewski and Janz, 2014 Nowlan et al., 2018 |






**Table 3: GeoTASO/GCAS Flight Summary for LISTOS**

| Flight | Date | Time (UTC fractional hour) | Pollution Scale (95th percentile ×10^15 molecules cm^-2) | % Cloudy Pixels | # Valid Pandora Coincidences | # Valid TROPOMI Coincidences | Flight pattern type (Fig. 1) |
|---|---|---|---|---|---|---|---|
| 1 | 18 Jun. 2018 | 12.0-15.6 | | | | | Large |
| 2 | | 17.0-20.7 | | | | | Large |
| 3 | 25 Jun. 2018 | 12.5-15.7 | 7.3 | 10 | 5 | 34 | Small |
| 4 | | 16.8-20.3 | 7.2 | 5 | | | Small |
| 5 | 30 Jun. 2018 | 12.2-15.6 | 11.2 | 0 | 9 | 65 | Small |
| 6 | | 16.7-20.4 | 13.5 | 1 | | | Small |
| 7 | 02 Jul. 2018 | 11.4-16.6 | 14.5 | 0 | 7 | 18 | Small |
| 8 | | 17.9-21.5 | 18.9 | 0 | | | Small |
| 9 | 19 Jul. 2018 | 11.4-15.3 | 17.9 | 0 | 11 | 47 | Large |
| 10 | | 16.9-20.9 | 32.4 | 0 | | | Large |
| 11 | 20 Jul. 2018 | 11.4-15.3 | 30.4 | 3 | 15 | 38 | Large |
| 12 | | 17.1-21.1 | 16.3 | 5 | | | Large |
| 13 | 05 Aug. 2018 | 12.5-16.5 | 15.5 | 1 | 15 | 0 | Large |
| 14 | | 17.8-22.3 | 10.2 | 5 | | | Large |
| 15 | 06 Aug. 2018 | 11.7-16.0 | 21.3 | 0 | 13 | 11 | Large |
| 16 | | 17.2-21.5 | 16.1 | 5 | | | Small |
| 17 | 15 Aug. 2018 | 11.2-15.5 | 12.4 | 0 | 17 | 52 | Large |
| 18 | | 17.0-21.6 | 9.8 | 5 | | | Large |
| 19 | 16 Aug. 2018 | 11.3-15.3 | 13.7 | 17 | 16 | 31 | Small |
| 20 | | 17.3-21.5 | 9.8 | 2 | | | Small |
| 21 | 24 Aug. 2018 | 10.9-15.3 | 14.7 | 0 | 18 | 32 | Large |
| 22 | | 16.6-21.0 | 37.8 | 4 | | | Large |
| 23 | 28 Aug. 2018 | 11.3-15.3 | 16.6 | 0 | 15 | 10 | Small |
| 24 | | 16.6-20.3 | 16.0 | 2 | | | Small |
| 25 | 29 Aug. 2018 | 11.2-15.1 | 16.8 | 0 | 17 | 17 | Small |
| 26 | | 16.6-20.8 | 14.0 | 3 | | | Small |
| 27 | 06 Sept. 2018 | 11.9-15.8 | 11.8 | 9 | 13 | 33 | Small |
| 28 | | 17.2-21.4 | 12.2 | 5 | | | Small |
| 29 | 03 Oct. 2018 | 12.3-16.7 | | | | | Small |
| 30 | | 18.2-21.8 | | | | | Small |
| 31 | 19 Oct. 2018 | 12.8-15.2 | | | | | Small |
| 32 | | 16.8-20.3 | | | | | Small |



⌐140

**Table 4: Pandora sites and time of operation. Shaded boxes represent the months of LISTOS.**

| Pandora Name | Latitude, Longitude | Months with Valid Data (number of measurement days per month) | | | | | | | | | |
|---|---|---|---|---|---|---|---|---|---|---|---|
| | | 2018 | | | | | | | 2019 | | |
| | | J | J | A | S | O | N | D | J | F | M |
| QueensNY | 40.7361, -73.8215 | 5 | 23 | 27 | 26 | 27 | 27 | 25 | 26 | 26 | 29 |
| BronxNY | 40.8679, -73.8781 | 6 | 29 | 29 | 16 | 21 | 10 | - | - | - | - |
| BayonneNJ | 40.6703, -74.1261 | - | 21 | 31 | 27 | 26 | 25 | 25 | 26 | 24 | 28 |
| FlaxPondNY | 40.9635, -73.1402 | 2 | 13 | 28 | 19 | 5 | - | - | - | - | - |
| WestportCT | 41.1183, -73.3367 | 5 | 19 | 29 | 25 | 27 | 24 | 26 | 23 | 5 | 22 |
| NewHavenCT | 41.3014, -72.9029 | 6 | 30 | 29 | 19 | 19 | 14 | 24 | 15 | - | - |
| RutgersNJ | 40.4622, -74.4294 | 2 | 30 | 30 | 21 | 27 | 22 | 25 | 21 | 5 | 21 |
| MadisonCT | 41.2568, -72.5533 | 7 | 13 | - | - | - | - | - | - | - | - |
| BranfordCT | 41.2420, -72.7604 | - | 9 | 30 | 4 | - | - | - | - | - | - |

⌐145

⌐150



**Table 5: Statistics for TROPOMI and airborne comparisons with the coincidence criteria of CRF < 50% and aircraft sampled within ± 30 minutes of the S5P overpass with different a priori profiles and indication of whether the $\Delta_{CS}$ threshold is applied.**

| TROPMI Dataset | $\Delta_{CS}$ < 50hPa | RMA Fit | $r^2$ | Median Percent Difference | N |
|---|---|---|---|---|---|
| Standard Slant Column | No | y=0.58x+1.5×10¹⁵ | 0.95 | -- | 621 |
| | Yes | y=0.59x+1.5×10¹⁵ | 0.96 | -- | 388 |
| Standard TrVC | No | y=0.71x+0.9×10¹⁵ | 0.90 | -11% | 621 |
| | Yes | y=0.68x+0.6×10¹⁵ | 0.96 | -19% | 388 |
| NAMCMAQ TrVC | No | y=0.84x+1.0×10¹⁵ | 0.83 | 4% | 621 |
| | Yes | y=0.77x+0.7×10¹⁵ | 0.95 | -7% | 388 |


**Table 6: Statistics between Pandora and TROPOMI by site for the LISTOS period as well as extended to 19 March 2019**

| | LISTOS Only (June-September 2018) | | | | | Valid data from June 2018-March 2019 | | | | |
|---|---|---|---|---|---|---|---|---|---|---|
| Site | RMA Fit | $r^2$ | Median % Difference | Median Column Difference | N | RMA Fit | $r^2$ | Median % Difference | Median Column Difference | N |
| QueensNY | Y=0.77x+0.6×10¹⁵ | 0.87 | -9% | -0.5×10¹⁵ | 22 | Y=0.63x+1.3×10¹⁵ | 0.76 | -23% | -2.1×10¹⁵ | 68 |
| BronxNY | Y=0.81x+0.03×10¹⁵ | 0.90 | -15% | -1.1×10¹⁵ | 20 | Y=0.73x+0.5×10¹⁵ | 0.87 | -15% | -1.1×10¹⁵ | 33 |
| BayonneNJ | Y=0.84x-2.1×10¹⁵ | 0.87 | -38% | -4.1×10¹⁵ | 9 | Y=0.74x-1.8×10¹⁵ | 0.88 | -41% | -5.3×10¹⁵ | 45 |
| WestportCT | Y=0.49x+1.1×10¹⁵ | 0.50 | -19% | -0.6×10¹⁵ | 21 | Y=0.68x+0.4×10¹⁵ | 0.95 | -21% | -0.9×10¹⁵ | 49 |
| RutgersNJ | Y=0.63x+0.4×10¹⁵ | 0.69 | -26% | -0.9×10¹⁵ | 6 | Y=0.76x-0.1×10¹⁵ | 0.95 | -24% | -1.4×10¹⁵ | 33 |
| FlaxPondNY | Y=0.53x+0.4×10¹⁵ | 0.59 | -37% | -1.7×10¹⁵ | 23 | Y=0.53x+0.5×10¹⁵ | 0.60 | -37% | -1.4×10¹⁵ | 25 |
| NewHavenCT | Y=0.52x-0.5×10¹⁵ | 0.29 | -52% | -2.7×10¹⁵ | 25 | Y=0.70x-1.3×10¹⁵ | 0.71 | -50% | -2.7×10¹⁵ | 47 |
| BranfordCT | Y=1.22x-2.7×10¹⁵ | 0.31 | -46% | -1.9×10¹⁵ | 22 | Y=1.2x-2.7×10¹⁵ | 0.31 | -46% | -1.9×10¹⁵ | 22 |
| MadisonCT | Y=1.94x-2.7×10¹⁵ | 0.12 | -24% | -0.6×10¹⁵ | 8 | Y=2.4x-3.9×10¹⁵ | 0.02 | -24% | -0.7×10¹⁵ | 11 |




**Table 7: Summary statistics for Pandora and TROPOMI over the LISTOS time period and extended to 19 March 2019 with different a priori profiles and indication of whether the ΔCS threshold is applied.**

| Time Period | Location | TROPMI Dataset | $\Delta_{CS}$ < 50hPa | RMA Fit | $r^2$ | Median Percent Difference | N |
|---|---|---|---|---|---|---|---|
| LISTOS Only | All Sites | Standard | No | $y=0.82x-0.6\times10^{15}$ | 0.79 | -30% | 294 |
| | | | Yes | $y=0.80x-0.7\times10^{15}$ | 0.84 | -33% | 156 |
| | | NAMCMAQ | No | $y=1.05x-0.7\times10^{15}$ | 0.77 | -9% | 294 |
| | | | Yes | $y=0.82x-0.2\times10^{15}$ | 0.80 | -19% | 156 |
| LISTOS Only<br>26 June 2018<br>—<br>19 March 2019 | RutgersNJ<br>BayonneNJ<br>QueensNY<br>WestportCT | Standard | No | $y=0.78x-0.5\times10^{15}$ | 0.79 | -17% | 132 |
| | | | Yes | $y=0.76x+0.1\times10^{15}$ | 0.88 | -19% | 58 |
| | | | No | $y=0.74x+0.2\times10^{15}$ | 0.82 | -21% | 373 |
| | | | Yes | $y=0.78x-0.3\times10^{15}$ | 0.87 | -27% | 195 |






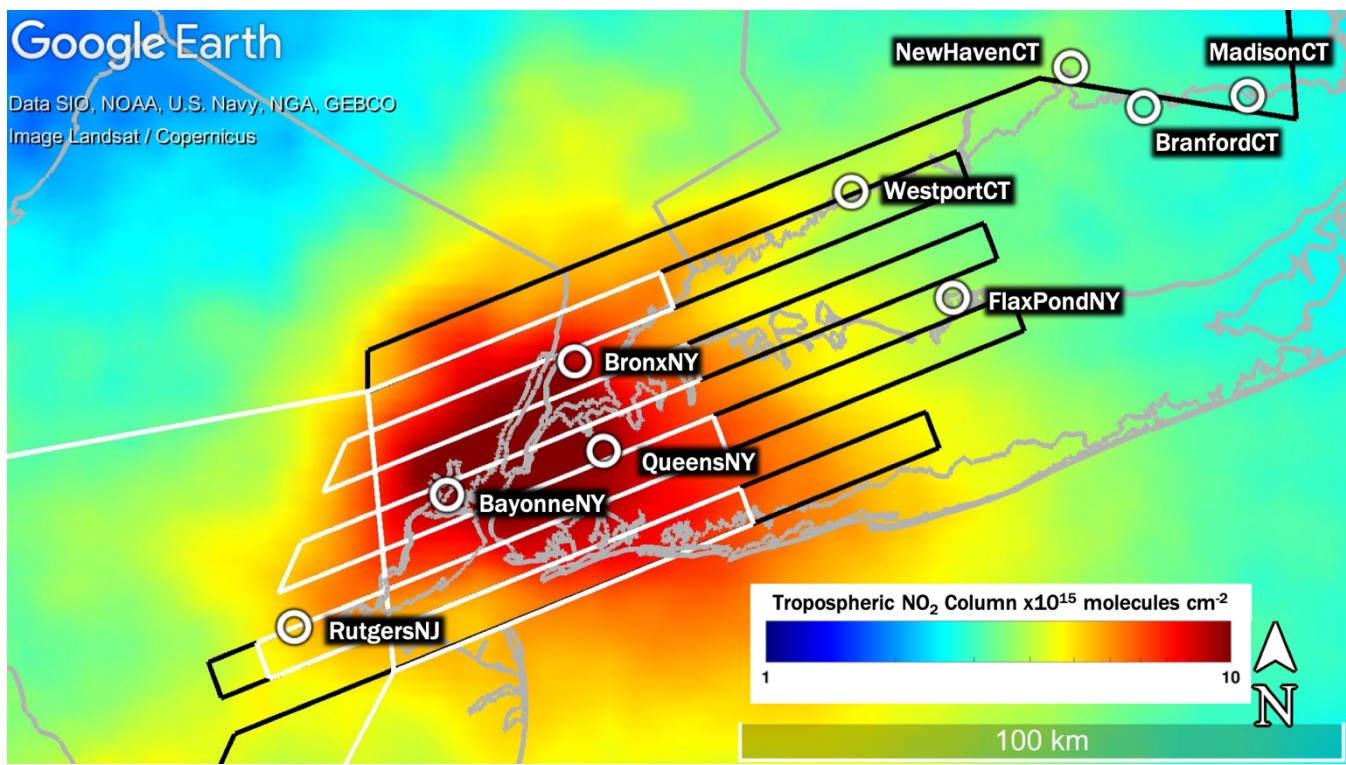

**Figure 1: Map showing the annual average TROPOMI tropospheric NO₂ columns between April 2018-March 2019. Overlaid circles show the locations of the nine Pandora spectrometer considered in this analysis. Table 4 shows when each of these instruments operated. The black and white lines represent the two types of flight plans flown by the airborne spectrometers (large in black and small in white). This map was created in © Google Earth Pro.**



**Figure 2: Scatter plots of the temporally closest Pandora TrVC to the aircraft overpass (± min/max observation within a ± 5-minute window from the aircraft overpass) vs. median airborne TrVC within a 750m radius of Pandora (±10[th]-90[th] percentile) with labeled statistics. 1:1 line is indicated with the grey dashed line. The solid black lines indicate the RMA linear regression for sites with r[2] greater than 0.5.**



**Figure 3: (a) Scatter plot showing the temporally closest Pandora TrVC to the aircraft overpass (± min/max observation within a ± 5-minute window from the aircraft overpass) vs. the median airborne TrVC (±10th-90th percentile) within a 750 m radius of the Pandora site. The thick solid black line represents the RMA linear regression. Each point is colored by Pandora location where the outlined squares are points where Pandora TrVCs do not vary more than 30% within a ± 15-minute window from the aircraft overpass, whereas the circles indicate times where Pandora TrVCs do vary more than 30%. (b) The difference between airborne and Pandora tropospheric NO₂ columns vs. time of day in hours (UTC) colored similarly to (a).**





**Figure 4:** Box plots (95-75-50-25-5) showing the airborne column (a) column difference and (b) percent difference from Pandora binned at the labeled thresholds ($\times 10^{15}$) as well as all data points (right). The number of points in each bin are indicated by the numbers in parentheses above the x-axis label.

**Figure 5: Maps demonstrating how airborne data is matched to TROPOMI for 3 out of 15 example overpasses: (top) VIIRS true color imagery (source: https://worldview.earthdata.nasa.gov/: last accessed 18 April 2020), (second row) overlaid TROPOMI TrVCs where CRFs < 50%, (third row) overlaid airborne data collected within ± 30 minutes of the TROPOMI overpass with outlined TROPOMI pixels with CRFs < 50% and area mapped by aircraft > 75%, (bottom) airborne NO$_2$ columns data scaled to the TROPOMI pixel. All maps were created in © Google Earth Pro.**

195

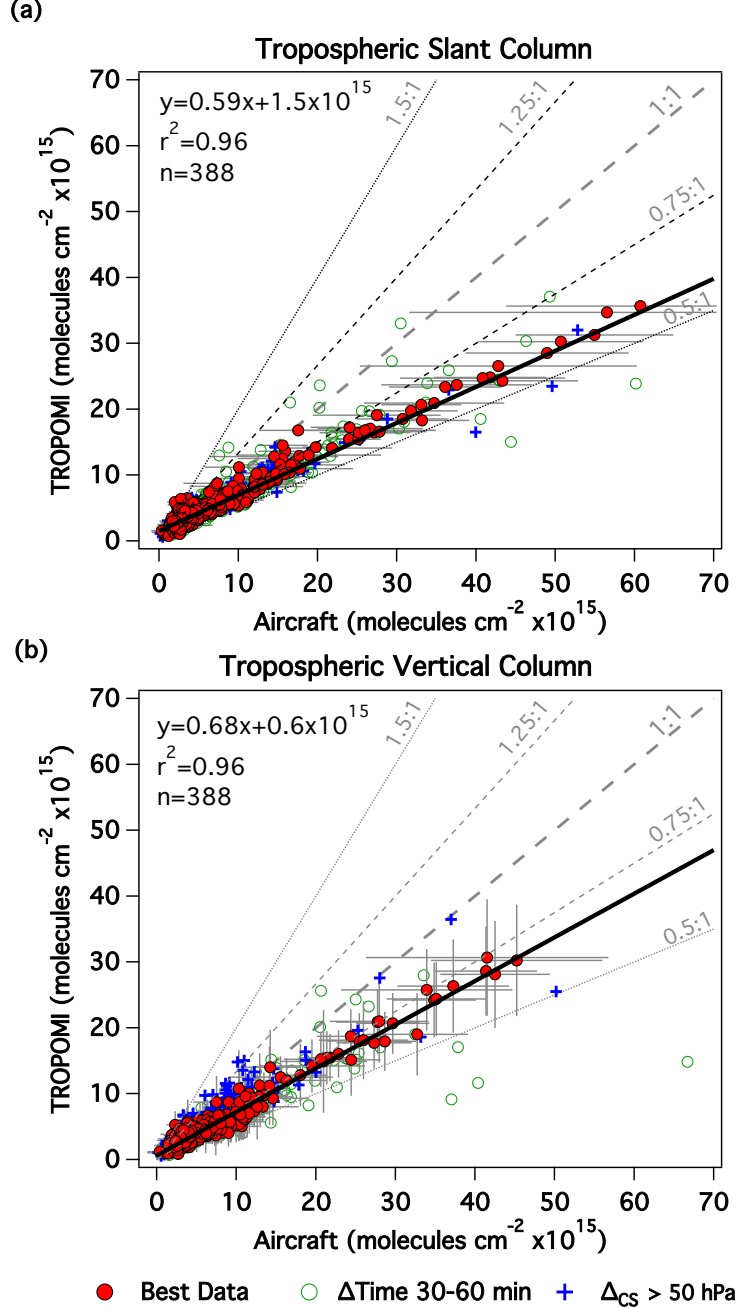

**Figure 6:** Scatter plots of airborne data gridded and scaled up to the TROPOMI pixel footprint vs. TROPOMI NO₂ tropospheric (a) slant column and (b) vertical column that are at least 75% mapped with a CRF < 50 % within ± 30 min of the TROPOMI overpass in red circles (open green circles show points when the time window is expanded to ± 60 min and blue crosses symbolize points where $\Delta_{CS} > 50$ hPa). The horizontal bars indicate the sub-pixel heterogeneity measured by the aircraft quantified as the standard deviation of aircraft slant columns over that pixel and vertical bars in (b) show the reported precision of the TROPOMI TrVC.



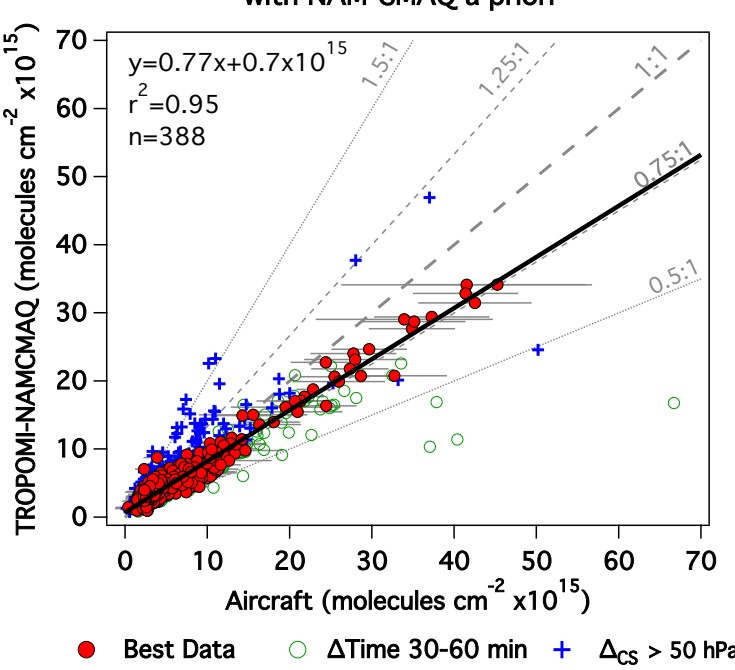

1205   **Figure 7: Scatter plots of airborne data gridded and scaled up to the TROPOMI pixel footprint vs. TROPOMI-NAMCMAQ NO₂**
**TrVCs that are at least 75% mapped with a CRF < 50 % within ± 30 min of the TROPOMI overpass in red circles (open green**
**circles show points when the time window is expanded to ± 60 min and blue crosses symbolize points where Δ_CS > 50 hPA). The**
**horizontal bars indicate the sub-pixel heterogeneity measured by the aircraft quantified as the standard deviation of aircraft vertical**
**columns over that TROPOMI pixel.**

1210





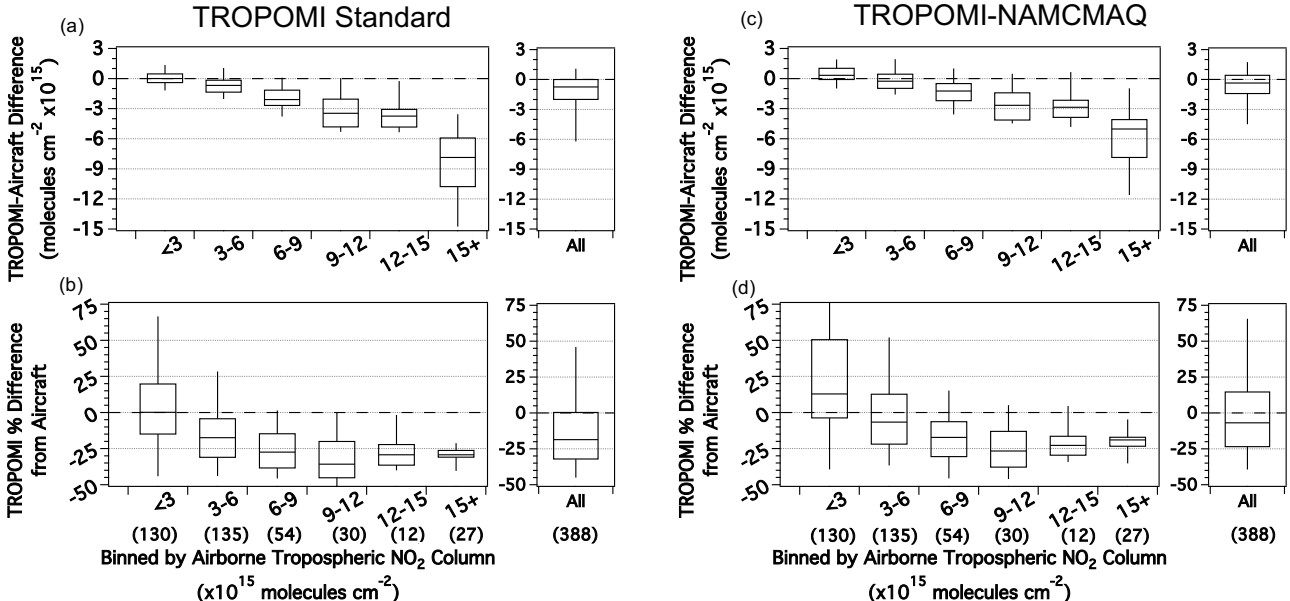

**Figure 8: Box plots (95-75-50-25-5) showing the TROPOMI TrVC (a) column difference and (b) percent difference from airborne TrVCs binned at the labeled thresholds (×10¹⁵) as well as for the total dataset (right), along with the equivalent box plots for TROPOMI-NAMCMAQ in (c) and (d). The number of points in each bin are indicated by the numbers in parentheses above the x-axis label.**

1215







**Figure 9: Scatter plots of the median Pandora TrVC within ± 30 min of the S5P overpass vs. TROPOMI TrVC for all coincidences with CRF < 50%, $\Delta_{CS}$ < 50 hPa between June 25$^{th}$ 2018 and 19 March 2019 at each individual site. Coincidences during the LISTOS intensive period (through the end of September 2018) are outlined in black. Vertical bars indicate the reported precision of TROPOMI TrVCs and the horizontal bars are the 10$^{th}$-90$^{th}$ percentile of Pandora TrVCs within ± 30 min of the S5P overpass. 1:1 line is indicated with the grey dashed line. Statistics are summarized in Table 6 but the RMA regression lines are shown for datasets with r$^2$ greater than 0.5 (solid black line is for the LISTOS timeframe and dashed black line is all data).**

1220







Figure 10: Scatter plot showing coincident (a) TROPOMI Standard TrVCs and (b) TROPOMI-NAMCMAQ TrVCs with CRF < 50% vs. median Pandora NO$_2$ TrVC over a ± 30-minute temporal window. Red points have a $\Delta_{CS}$ < 50 hPa, whereas blue crosses have a $\Delta_{CS}$ > 50 hPa. The horizontal bars represent the 10$^{th}$-90$^{th}$ percentile of Pandora data within the ±30 min temporal window. The vertical bars in (a) represent the reported precision of TROPOMI Standard. The thick solid black line represents the RMA linear regression applied to the red data points. The box plots (95-75-50-25-5) show the TROPOMI TrVC percent difference from Pandora for the red data points to the right of each scatter plot.



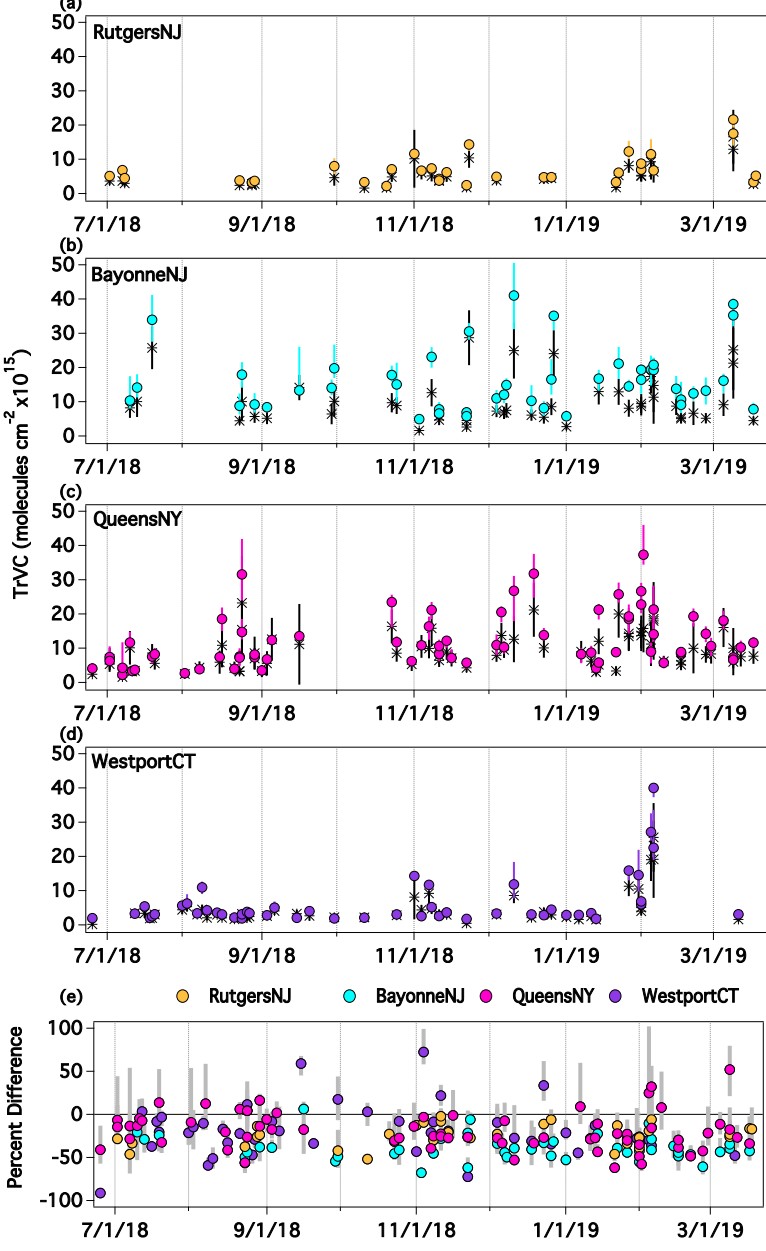

**Figure 11: Time series of Pandora and TROPOMI Standard TrVCs from 25 June 2018 through 19 March 2019. Circles represent the Pandora data ± 10th-90th percentile in the ± 30-minute window and the stars indicated the TROPOMI TrVC ± the reported precision at (a) RutgersNJ, (b) BayonneNJ, (c) QueensNY, and (d) WestportCT. The percent difference of the TROPOMI Standard TrVC from Pandora colored by site is shown in (e) and the grey bars indicate the 10th-90th percentile of the column difference of TROPOMI TrVC from the sub-temporal Pandora data.**

1235



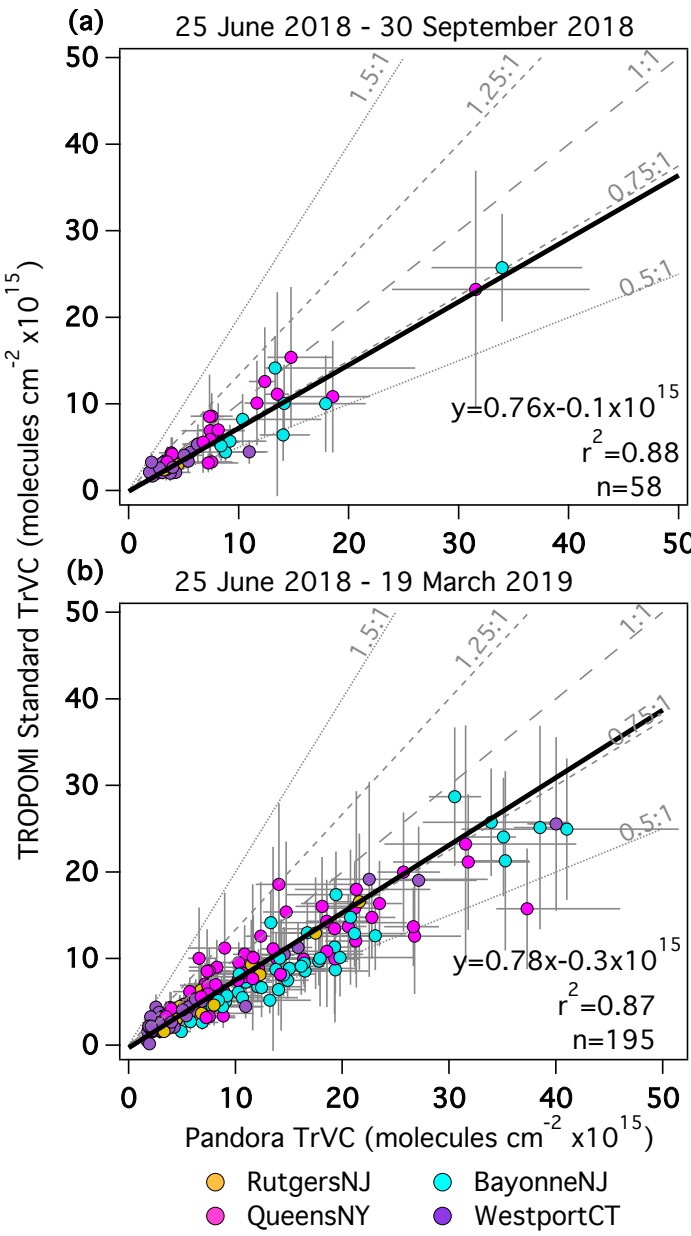

**Figure 12: TROPOMI Standard vs Pandora TrVCs colored by site during (a) the LISTOS intensive period and (b) coincidences**
**extending from 25 June 2018 – 19 March 2019. The horizontal bars represent the 10th-90th percentile of Pandora data within the ±30**
**min temporal window. The vertical bars represent the reported precision of TROPOMI. Each point is colored by Pandora location.**