# Peer review of "Evaluating Sentinel-5P TROPOMI tropospheric NO2 column densities with airborne and Pandora spectrometers near New York City and Long Island Sound"

_Atmospheric Measurement Techniques, 2020_

## Referee Comment (RC1) · Anonymous Referee #3 · 17 Jun 2020

This paper presents a comprehensive evaluation of the TROPOMI satellite NO2 product v1.2 for the New York City/Long Island Sound region, where the NO2 TrVC has high spatial and temporal heterogeneity. The NO2 TrVC measurements from both airborne and ground-based Pandora spectrometers are used to compare with the TROPOMI NO2 products. While Pandora spectrometers provided continuous long-term measurements, airborne spectrometers provide observations with more spatially representative of the satellite measurements. The effects of the cloud retrieval and a priori profile on the biases in TROPOMI NO2 product are analyzed. The study is interesting and

provides a welcome addition to the literatures on the measurements of the NO2 TrVCs from satellite, airborne and ground-based spectrometers. The manuscript is well written and the presentation looks good. I would recommend acceptance for publication after the following comments have been addressed.

Specific comments:

What are the exposure time for each scan of GeoTASO and GCAS during the flight and corresponding distance the airplanes flied? I did not find this information in Sect. 2.2.

The definitions of the tropospheric column seem to be different for satellite (L141-142), airborne (L229-230), and ground-based measurements. In other wards, different 'tropopause altitudes' are used to derive the TrVCs of NO2. Considering that NO2 concentrations in the upper troposphere near the tropopause may be sufficiently large, could these differences in the definition affect the comparisons among the three data sets? How is the airborne stratospheric columns of NO2 retrieved (L268-269)?

L78-79: In addition to the gradient-smoothing effect, the aerosol-shielding effect may also make a contribution to the uncertainties in the validation of satellite products by ground-based spectrometer, particularly in high-aerosol-load areas (e.g., Ma et al., 2013;Jin et al., 2016). How about the typical aerosol levels over the investigated region? Can the aerosol shielding effect be large enough to affect the comparison of Pandora with TROPOMI and airborne spectrometer measurements?

Technical issues:

L21-23: please rephrase the first sentence in the Abstract. It should be stated that the measurements were made or the measurement data were collected. Better to describe more clearly which coincided with the early measurements from the Sentinel-5P TROPOMI instrument?

L37: change 'biggest' to 'largest'.

L124: the words 'to be' can be deleted.

L153: how is the qa_value defined?

L163: what does the dynamic range of NO2 refers to?

L74: please check the phrase 'through June 30'. Did Hu25 fly only one day?

L187: please give the pressure altitude in hPa.

L790; 'This is the first work that airborne spectrometer measurement dataset has been used to . . .'?

Figure 2: please add x10ˆ(15) to the labels of both x-axes and y-axes.

References

Ma, J. Z., Beirle, S., Jin, J. L., Shaiganfar, R., Yan, P., and Wagner, T.: Tropospheric NO2 vertical column densities over Beijing: results of the first three years of ground-based MAX-DOAS measurements (2008-2011) and satellite validation, Atmos. Chem. Phys., 13, 1547-1567, 10.5194/acp-13-1547-2013, 2013.

Jin, J., Ma, J., Lin, W., Zhao, H., Shaiganfar, R., Beirle, S., and Wagner, T.: MAX-DOAS measurements and satellite validation of tropospheric NO2 and SO2 vertical column densities at a rural site of North China, Atmospheric Environment, 133, 12-25, http://dx.doi.org/10.1016/j.atmosenv.2016.03.031, 2016.

---

## Referee Comment (RC2) · Anonymous Referee #1 · 18 Jun 2020

The present manuscript presents the evaluation of S5P TROPOMI tropospheric NO2 column densities with the aid of airborne and ground-based spectrometers in New York City and Long Island Sound. The advantage/ challenge of this region is that the NO2 concentrations are highly heterogeneous in time and space. The validation of S5P TROPOMI tropospheric NO2 column densities is separated in two major categories: (1) comparison between airborne NO2 TrVC and TROPOMI NO2 TrVC and (2) comparison between ground-based NO2 TrVC and TROPOMI NO2 TrVC. From the above-mentioned comparisons, the authors observe a bias in TROPOMI NO2 TrVC and the

effect of clouds and a-priori profile in the TROPOMI retrieval are examined into details. I strongly recommend the publication of the manuscript after consideration of a minor number of specific considerations: Specific comments:

– Page 2, Line 60: It would be interesting to add the exact spatial resolution of OMI and OMPS

– Page 4, Line 110: I suggest that for the reader it would be more practical if you include a small separate section or subsection called "LISTOS campaign" and write there the information about the campaign, as you already did in Section 2.

– Page 7, Line 218: Please explain the PRATMO acronym

– Page 8, Line 234: If I understand well, did you assume that the aerosol a-priori profile in the AMF calculation is zero? So, you assumed that no aerosols are present in the atmosphere, or not? If this the case, is this assumption leading to realistic results?

– Page 18, Line 580: Can you provide an approximate value for the Pandora horizontal sensitivity?

– Page 19, Line 581: This means that the Pandora data are not filtered for clouds?

– Page 21, Line 650: Is there a reason why you did not compare Pandora TrVC (vs) TROPOMI-NAMCMAQ for the extended time period? I would be interesting to add a figure with this comparison.

– Page 21, Line 678: You could cite studies that use MAX-DOAS measurements to evaluate the TROPOMI NO2 product.

– Page 22, Line 699: Please add some reference studies.

– Page 39, Table 5: Is there a reason why you did not present the median percentage difference for the Standard Slant Column?

– Page 46, Figure 6: I would suggest that in Fig. 6a, you include the reported

TROPOMI SCD error.

– Page 47, Figure 7: The figure does not contain error bars in the vertical axis. Is there any way to estimate the TROPOMI-NAMCMAQ error and add it to the figure?

---

## Referee Comment (RC3) · Anonymous Referee #2 · 19 Jun 2020

This paper by Judd et al. compares satellite-based TROPOMI tropospheric NO$_2$ measurements with airborne- and ground-based Pandora measurements in the New-York City/Long Island Sound region. It contributes to the Sentinel-5P TROPOMI validation and is the first validation paper for the new satellite instrument with airborne campaign measurements which have a more spatially representativity than ground-based measurements. In addition, long-terme ground-based Pandora measurements are used and compared to the airborne and satellite based NO$_2$ measurements. The strength of both reference measurements are used to evaluate TROPOMI tropospheric NO$_2$ col-

umn densities.

The evaluation found a low bias of the TROPOMI tropospheric vertical column (TrVC) compared to Pandora and aircraft tropospheric vertical column, more pronounced for aircraft than Pandora measurements. Although using a higher resolution a priori vertical profile for the TROPOMI data improves the low bias, there is still a low bias, especially for more polluted cases and further investigations are needed in future studies.

Cloud retrieval effects are discussed. A new quality criterion was introduced which excludes pixel where the difference between retrieved cloud pressure and surface pressure exceeds 50 hPa to exclude pixels where cloud shielding occurred over cloud free scenes. These pixels compensate partially for the TROPOMI TrVC low bias but lower the correlations with reference measurements.

The paper is well written and of significance for the validation of the new satellite Sentinel-5P TROPOMI tropospheric $NO_2$ measurements. Therefore, I recommend publication in AMT with minor revisions.

Specific comments:

Line 197: "All reference spectra were co-located with total column $NO_2$ measurements from Pandora spectrometers: $5.6*10^{15}$ molecules cm$^{-2}$ at MadisonCT on June 30th, $5.7*10^{15}$ molecules cm$^{-2}$ at MadisonCT on July 2nd, and $6.2*10^{15}$ molecules cm$^{-2}$ at WestportCT on August 5th, with values estimated to be over 50% stratospheric."
What is done with the collocated Pandora measurements? How is the 50% stratospheric estimated?

Line 292: What is the spatial coincidence criterion for Pandora comparisons to TROPOMI? Is it the nearest pixel, a mean, is the viewing direction considered?

Line 254: "All Pandora data are converted from total vertical columns to TrVCs by subtracting either the airborne or TROPOMI retrieved stratospheric columns for comparison purposes."
Is the Pandora converted with TROPOMI retrieved stratospheric column used for

TROPOMI comparisons and Pandora converted with airborne for airborne comparisons? How is the airborne stratospheric column retrieved?

Line 450: Why was this feature only seen by this excluded Pandora?

Line 571: Is there an explanation why the slope is much better and the correlation much worse when comparing TROPOMI and Pandora instead of TROPOMI and aircraft measurements?

Line 675: Lorente et al. did not used Pandora spectrometers for validation, they also found a low bias but with in-situ measurements.

Technical corrections:

Line 99: LISTOS is defined and used already in line 21 and 36.

Line 283: "TROPOMI NO$_2$ columns"
Better TrVC to be consistent to the other TrVCs in the sentence.

Line 372: "(Table S1, compare Row I to Row B) slightly improves the correlation (r$^2$ increases from 0.93 to 0.94)"
Row I is 0.94 and row B 0.92. Value 0.93 should be changed to 0.92 and order of "compare Row I to Row B" should be changed to "B to I" to make it consistent to the values order.

Line 420: "with large sub-pixel variation as indicated by the horizontal whiskers in the plot" There is a better explanation but some lines later (Line 433). This one could be replaced by the later one.
"The horizontal bars in Fig. 6 show the standard deviation of the subpixel airborne TrVCs within each TROPOMI pixel."

Line 556/Figure 9: Statistics are only listed in the table. It would be helpful for a better reading to have at least the statistics of the LISTOS time period data in the figure especially because these are much more discussed in the following than the statistics

of the long-term TROPOMI-Pandora comparison.

Line 651: $r^2$ of 0.89 should be 0.88 corresponding to the figure.

Line 714: $r^2$ of 0.84 should be 0.88

Line 722: and

Table 2: kg instead of lbs

Table 3: Short explanation for shaded boxes

Figure 1: Nine Pandora spectrometers instead of spectrometer.

Figure 2: Explanation to horizontal and vertical bars with "variability at the time of measurement" is missing in figure caption.

Figure 10: The period (LISTOS or extended long-term) of the used data is missing.

Figure 12: (a) Also for the LISTOS comparison only the extended stations are used

Supplement Line 53: "to remove the estimated of loss of sensitivity"
First "of" can be removed

---

## Author Comment (AC1) · 13 Aug 2020

*Reviewer 1:*
*We would like to thank you for your comments as we appreciate the time dedicated for this review and have made changes to the manuscript to reflect the suggestions. As a note: I noticed that these comments are based on the initial submission before technical edits and not the version posted for discussion, therefore some suggestions were not as clear but were addressed to the best of my ability. Individual comments from the review are bolded with our responses in italics.*

**This paper presents a comprehensive evaluation of the TROPOMI satellite NO2 product**

**v1.2 for the New York City/Long Island Sound region, where the NO2 TrVC has high**

**spatial and temporal heterogeneity. The NO2 TrVC measurements from both airborne**

**and ground-based Pandora spectrometers are used to compare with the TROPOMI**

**NO2 products. While Pandora spectrometers provided continuous long-term measurements,**

**airborne spectrometers provide observations with more spatially representative**

**of the satellite measurements. The effects of the cloud retrieval and a priori profile**

**on the biases in TROPOMI NO2 product are analyzed. The study is interesting and**

**provides a welcome addition to the literatures on the measurements of the NO2 TrVCs**

**from satellite, airborne and ground-based spectrometers. The manuscript is well written**

**and the presentation looks good. I would recommend acceptance for publication**

**after the following comments have been addressed.**

**Specific comments:**

**What are the exposure time for each scan of GeoTASO and GCAS during the flight and**

**corresponding distance the airplanes flied? I did not find this information in Sect. 2.2.**

*Integration times for GeoTASO are fixed at 250 ms and GCAS integration times span 225 ms to 750 ms (most the time at 225 ms). I added this information to Table 2. I also added the median aircraft speeds in the text. The ground speed of the HU-25 at altitude averaged 215 m/s, therefore each exposure was ~ 53 m in distance along track. The King Air is slower than the HU-25 with an average ground speed 123 m/s. With the range of integration times, this would mean that one exposure ranged from 30-90 m. Multiple images are co-added to create a pixel size close to 250 m.*

**The definitions of the tropospheric column seem to be different for satellite (L141-**

**142), airborne (L229-230), and ground-based measurements. In other wards, different**

**'tropopause altitudes' are used to derive the TrVCs of NO2. Considering that NO2**

**concentrations in the upper troposphere near the tropopause may be sufficiently large,**

**could these differences in the definition affect the comparisons among the three data**

**sets? How is the airborne stratospheric columns of NO2 retrieved (L268-269)?**

*Starting with the last question, as another review ask about this this as well. We clarified the airborne spectrometer stratospheric estimation. The airborne stratospheric component is estimated using a stratospheric NO2 climatology developed using the PRATMO (PRather ATmospheric MOdel) Photochemical Box Model (Prather, 1992; McLinden et al., 2000; Nowlan et al., 2016). The PRATMO column is bias corrected daily using*

*TROPOMI NO$_2$ stratospheric vertical columns by calculating the average offset between the two datasets over the LISTOS domain for each day (ranging from 5x10$^{13}$ to 6x10$^{14}$ molecules cm$^{-2}$). This is stated in Section 2.3. I also avoided calling it stratospheric 'retrieved' as the actual stratospheric vertical column component is estimated from outside data (TROPOMI+PRATMO) and not directly retrieved but rather that signal is removed when doing the differential slant column to tropospheric vertical column conversion.  I added text to Sect. 2.3 to more clearly state this conversion:*

> *"Differential slant columns are converted to below aircraft vertical columns (assumed as the tropospheric vertical column, TrVC) by subtracting the estimated stratospheric slant column (climatology bias corrected daily with TROPOMI multiplied by the stratospheric AMF), adding the estimated reference slant column amount (from Pandora) and dividing by the tropospheric air mass factor, similar to Eq. 1 in Judd et al. (2019) or Eq 4. in Nowlan et al. (2018)."*

*For each dataset comparisons, we aimed to keep the stratospheric component compatible between the reference and the evaluated measurements.  These details are found within the manuscript.*

1. *Pandora v. Aircraft: The estimated aircraft stratospheric column is subtracted from Pandora. The uncertainty in that assumed value is in both datasets.*
2. *Pandora v. TROPOMI: For these comparisons, the stratospheric column from the TROPOMI product is subtracted from Pandora. Therefore, that assumption is made in both datasets.*
3. *Aircraft v. TROPOMI: Stratospheric columns retrieved from TROPOMI is part of the estimated airborne column and the largest errors would likely be associated with times furthest from the TROPOMI overpass time as the slope change throughout the day is estimated from the climatology created from PRATMO.  So, for airborne/TROPOMI comparisons during the time of the TROPOMI overpass time are mostly comparable.  They do use different definitions for the 'tropopause' however, if there were a significant feature making a difference then we would expect to see day-by-day clustering in the comparisons, which we do not. I did go back and calculate what our 'a priori' profile is between the aircraft and the TROPOMI tropopause pressure and that value is less than 2x10$^{14}$ molecules cm$^{-2}$ (median is 1.6x10$^{14}$ molecules cm$^{-2}$). We expect any impact to be minimal and would not affect the conclusions made about these comparisons.*

**L78-79: In addition to the gradient-smoothing effect, the aerosol-shielding effect may**

**also make a contribution to the uncertainties in the validation of satellite products by**

**ground-based spectrometer, particularly in high-aerosol-load areas (e.g., Ma et al.,**

**2013;Jin et al., 2016). How about the typical aerosol levels over the investigated region?**

**Can the aerosol shielding effect be large enough to affect the comparison of**

**Pandora with TROPOMI and airborne spectrometer measurements?**

*During Pandora+airborne spectrometer comparisons, over 90% of the coincidences have an AOT at 532 nm < 0.3 (measured by the co-located airborne lidar, HALO),  two coincidences are above 0.5 with a max of 0.7.  In the supplement, Figure S1 shows the comparison colored by AOT.  We added text with the details about AOT during these coincidences to give readers a gauge on aerosol loading. We have also discussed aerosol impacts during outlier coincidences as possible causes during individual cases, though did not find strong evidence that they were a regular impact.*

*We do not have regular AOT measurements to coincide with the Pandora sites for Pandora/TROPOMI coincidences, therefore we rely on the Pandora algorithm to filter out scenes that are extremely aerosol polluted.  In this work, we only use direct-sun measurements, so most of the signal measured by the spectrometer in clear scenes is from the direct solar beam.  In the presence of clouds or heavy aerosols, scattered light can become a small fraction of the light observed by the Pandora direct-sun measurement and in the case of heavy aerosol loading (~AOT>1), these measurements are flagged as lower quality (like a cloud). This work only includes high quality measurements, therefore, cases of extreme aerosols are inherently filtered out.*

*In scenes with lighter aerosol loading, there is the potential for aerosol impacts to the TROPOMI retrieval. In Sect 4.1 we mention that scattering from aerosols are assumed as indirectly sensed through the cloud retrieval, though it is not explicitly accounted for in the TROPOMI retrieval. During aircraft coincidences with TROPOMI, AOT at 532nm measured by HALO has a mean of 0.22 and a standard deviation of 0.15. This detail been added to Sec. 4.1 for additional context on aerosol loading. In future work, we plan to use aerosol profile measurements from HALO to estimate the sensitivity to aerosol loading in this region, which would enable us to more explicitly answer this question.*

*For now, we have added the following text to the paper in Sect 7. that references sources promote that aerosol impacts should be included in future investigations*

> *"One component not explicitly explored in this work, that should be in the future, is the potential impact of aerosols on the TROPOMI retrieval and whether their indirect accounting through the cloud retrieval accurately reflects the impacts within the radiative transfer calculations for the air mass factor calculation (e.g., Leitão et al., 2010; Ma et al., 2013; Jin et al., 2016)."*

**Technical issues:**

**L21-23: please rephrase the first sentence in the Abstract. It should be stated that**

**the measurements were made or the measurement data were collected. Better to**

**describe more clearly which coincided with the early measurements from the Sentinel-**

**5P TROPOMI instrument?**

*I reworded the first sentence to say: ' Airborne and ground-based Pandora spectrometer $NO_2$ column measurements were collected during the 2018 Long Island Sound Tropospheric Ozone Study (LISTOS) in the New York City/Long Island Sound region which coincided with early observations from the Sentinel-5P TROPOMI instrument.'*

**L37: change 'biggest' to 'largest'.**

*Change made as requested.*

**L124: the words 'to be' can be deleted.**

*I changed the phrase: 'to span late June through..'*

**L153: how is the qa_value defined?**

*QA_values are defined within the TROPOMI product file. This is not a value that I am defining. Information on how it is defined is located in the references in this part of the discussion (particularly the product user's manual; Eskes et al., 2019).*

**L163: what does the dynamic range of NO2 refers to?**

*The dynamic range is referring to the range of $NO_2$ columns observed from day to day. This can vary day to day from very clean (less than $1x10^{15}$) to very polluted (up to $100x10^{15}$). The main point in this discussion is that the peak in the annual average in that area is $12x10^{15}$, but day to day variations can be quite a bit more or less polluted than that.*

**L74: please check the phrase 'through June 30'. Did Hu25 fly only one day?**

*Table 3 has a summary of the flights and 'through June $30^{th}$ was referring to all flights prior to and on June $30^{th}$. I changed the phrasing to say 'GeoTASO was flown on the NASA LaRC HU-25 Falcon during the three June flight days…'*

**L187: please give the pressure altitude in hPa.**

*Instead, I removed the word pressure and refer to it as aircraft indicated altitude, as I am referring to the aircraft being set to fly at an altitude of 28,000 ft according to its altimeter.*

**L790; 'This is the first work that airborne spectrometer measurement dataset has been**

**used to . . .'?**

*Changed the sentence to say 'This is the first work that uses an airborne spectrometer dataset to evaluate the TROPOMI tropospheric NO$_2$ product.'*

**Figure 2: please add x10^(15) to the labels of both x-axes and y-axes.**

*This label is on both axes.*

**References**

**Ma, J. Z., Beirle, S., Jin, J. L., Shaiganfar, R., Yan, P., and Wagner, T.: Tropospheric NO2 vertical column densities over Beijing: results of the first three years of groundbased MAX-DOAS measurements (2008-2011) and satellite validation, Atmos. Chem. Phys., 13, 1547-1567, 10.5194/acp-13-1547-2013, 2013.**

**Jin, J., Ma, J., Lin, W., Zhao, H., Shaiganfar, R., Beirle, S., and Wagner, T.: MAXDOAS measurements and satellite validation of tropospheric NO2 and SO2 vertical column densities at a rural site of North China, Atmospheric Environment, 133, 12-25, http://dx.doi.org/10.1016/j.atmosenv.2016.03.031, 2016.**

---

## Author Comment (AC2) · 13 Aug 2020

*Reviewer 3:*

*We would like to thank you for your comments as we appreciate the time dedicated for this review and have made changes to the manuscript to reflect the suggestions. Individual comments from the review are bolded below with our responses in italics.*

**This paper by Judd et al. compares satellite-based TROPOMI tropospheric NO2 measurements with airborne- and ground-based Pandora measurements in the New-York City/Long Island Sound region. It contributes to the Sentinel-5P TROPOMI validation and is the first validation paper for the new satellite instrument with airborne campaign measurements which have a more spatially representativity than ground-based measurements. In addition, long-terme ground-based Pandora measurements are used and compared to the airborne and satellite based NO2 measurements. The strength of both reference measurements are used to evaluate TROPOMI tropospheric NO2 column densities. The evaluation found a low bias of the TROPOMI tropospheric vertical column (TrVC) compared to Pandora and aircraft tropospheric vertical column, more pronounced for aircraft than Pandora measurements. Although using a higher resolution a priori vertical profile for the TROPOMI data improves the low bias, there is still a low bias, especially for more polluted cases and further investigations are needed in future studies. Cloud retrieval effects are discussed. A new quality criterion was introduced which excludes pixel where the difference between retrieved cloud pressure and surface pressure exceeds 50 hPa to exclude pixels where cloud shielding occurred over cloud free scenes. These pixels compensate partially for the TROPOMI TrVC low bias but lower the correlations with reference measurements.**
**The paper is well written and of significance for the validation of the new satellite Sentinel-5P TROPOMI tropospheric NO2 measurements. Therefore, I recommend publication in AMT with minor revisions.**

**Specific comments:**
**Line 197: "All reference spectra were co-located with total column NO2 measurements from Pandora spectrometers: 5.6*10$_{15}$ molecules cm$_{-2}$ at MadisonCT on June 30th, 5.7*10$_{15}$ molecules cm$_{-2}$ at MadisonCT on July 2nd, and 6.2*10$_{15}$ molecules cm$_{-2}$ at WestportCT on August 5th, with values estimated to be over 50% stratospheric."**
**What is done with the collocated Pandora measurements? How is the 50% stratospheric estimated?**
*The Pandora measurements are collected during co-located reference spectra scenes by the airborne spectrometer and are used to estimate the total column for our reference. The airborne spectrometer NO2 data has its own above aircraft (stratospheric) value estimated based on coincident TROPOMI stratospheric columns with the diurnal pattern from a climatology created with the PRATMO photochemical box model. That separation is needed in the calculation converting differential slant column to vertical column (e.g., the screenshot equation from Lamsal et al., 2017) with various other versions found in Judd et al, (2019) and Nowlan et al. (2018). I also added some text to reflect this calculation in Sect 2.3.*

$$\Omega_v\downarrow = \frac{d\Omega_s - \Omega_v\uparrow \times A\uparrow + \left(\Omega_v{}^R\downarrow \times A^R\downarrow + \Omega_v{}^R\uparrow \times A^R\uparrow\right)}{A\downarrow}.$$

*They are estimated as 50% by using the estimated airborne stratospheric column.*

$$\Omega_{Pandora\ troposphere} = \Omega_{Pandora} - \Omega_{airborne\ stratosphere}$$
$$where\ \Omega_{Pandora\ troposphere} = \Omega_v^R \Downarrow \quad and\ \Omega_{airborne\ stratosphere} = \Omega_v^R \Uparrow$$

**Line 292: What is the spatial coincidence criterion for Pandora comparisons to TROPOMI? Is it the nearest pixel, a mean, is the viewing direction considered?**
*It is the pixel in which the Pandora resides and viewing direction is not considered. This is clarified in Sect. 2.5.*

**Line 254: "All Pandora data are converted from total vertical columns to TrVCs by subtracting either the airborne or TROPOMI retrieved stratospheric columns for comparison purposes."**

**Is the Pandora converted with TROPOMI retrieved stratospheric column used for TROPOMI comparisons and Pandora converted with airborne for airborne comparisons?**
*Yes and yes.*

**How is the airborne stratospheric column retrieved?**
*The airborne stratospheric component is estimated using a stratospheric $NO_2$ climatology developed using the PRATMO (PRather ATmospheric MOdel) Photochemical Box Model (Prather, 1992; McLinden et al., 2000; Nowlan et al., 2016). The stratospheric column is bias corrected daily using TROPOMI $NO_2$ stratospheric vertical columns by calculating the average offset between the two datasets over the LISTOS domain for each day (ranging from $5x10^{13}$ to $6x10^{14}$ molecules $cm^{-2}$). To reflect this I reworded the sentence to say: 'All Pandora data are converted from total vertical columns to TrVCs by subtracting either the airborne estimated or TROPOMI retrieved stratospheric columns for comparison purposes.'*

**Line 450: Why was this feature only seen by this excluded Pandora?**
*It is by the nature of the plume, which is stated in Section 4, which we did observe with the airborne spectrometer. See below for an image of the large change in morning and afternoon and the structure of the plume i*

[Figure]

Airborne spectrometer NO2 TrVC during the morning (top) and afternoon (bottom) flights of July 2nd, 2018. The labeled circles indicate 3 Pandora locations, where CCNY is not included in this analysis (will be investigated for future work).

**Line 571: Is there an explanation why the slope is much better and the correlation much worse when comparing TROPOMI and Pandora instead of TROPOMI and aircraft measurements?**
*The correlation is better for the airborne spectrometer because the TROPOMI sub-pixel variation is sampled by the airborne data. However, Pandora data is still subject to impacts from sub-pixel variability in this heterogeneous environment. It seems to be that the slope of the fit is also related to the sub-pixel heterogeneity. If you look at Table 6, you'll note that for sites where $r^2$ is greater than 0.5, the slopes range from 0.53 to 0.84.*

**Line 675: Lorente et al. did not used Pandora spectrometers for validation, they also found a low bias but with in-situ measurements.**
*You are right, the wording was wrongly reflecting Lorente et al. since they use in situ data. I broadened the scope of this paragraph now to include some other studies (including MAX-DOAS observational studies) as suggested by another reviewer.*

**Technical corrections:**
**Line 99: LISTOS is defined and used already in line 21 and 36.**
*I removed this definition of LISTOS. Now it should only be defined once in the abstract and once in the main manuscript.*

**Line 283: "TROPOMI NO2 columns"**
**Better TrVC to be consistent to the other TrVCs in the sentence.**
*I see this point. However, during editing, this sentence had already changed in the next version and doesn't have this discrepancy anymore.*

**Line 372: "(Table S1, compare Row I to Row B) slightly improves the correlation (r2 increases from 0.93 to 0.94)"**
**Row I is 0.94 and row B 0.92. Value 0.93 should be changed to 0.92 and order of "compare Row I to Row B" should be changed to "B to I" to make it consistent to the values order.**
*Thanks for catching the translation error in the $r^2$. This was fixed.*

**Line 420: "with large sub-pixel variation as indicated by the horizontal whiskers in the plot" There is a better explanation but some lines later (Line 433). This one could be replaced by the later one.**
**"The horizontal bars in Fig. 6 show the standard deviation of the subpixel airborne TrVCs within each TROPOMI pixel."**
*I removed the former phrase opting for the later discussion of spatial variation.*

**Line 556/Figure 9: Statistics are only listed in the table. It would be helpful for a better reading to have at least the statistics of the LISTOS time period data in the figure especially because these are much more discussed in the following than the statistics of the long-term TROPOMI-Pandora comparison.**
*I added the statistics for the LISTOS time period to Figure 9.*

**Line 651: r2 of 0.89 should be 0.88 corresponding to the figure.**
*Updated the figure. $R^2$ is 0.885, which rounds up to 0.89. Thanks for catching this error in translation.*

**Line 714: r2 of 0.84 should be 0.88**
*It should be 0.84 as I am referring to only the LISTOS timeframe for all Pandora sites (Figure 10a).*

**Line 722: and**
*corrected*

**Table 2: kg instead of lbs**
*Converted the weights to kg.*

**Table 3: Short explanation for shaded boxes**
*Added: 'Flights with shaded boxes are not considered in this analysis.' to the table description. Explanation is in the text as to why they aren't include: 'Only flights from 25 June – 6 September (13 flight days) are considered in this analysis due to availability of the high-resolution model data used to provide the a priori $NO_2$ profile shapes in the full vertical column retrieval (Table 1)'*

**Figure 1: Nine Pandora spectrometers instead of spectrometer.**
*Fixed*

**Figure 2: Explanation to horizontal and vertical bars with "variability at the time of measurement" is missing in figure caption.**
*For brevity it is described by the '($\pm$ min/max observation within a $\pm$ 5-minute window from the aircraft overpass)' and '($\pm 10^{th}$-$90^{th}$ percentile)'.*

**Figure 10: The period (LISTOS or extended long-term) of the used data is missing.**
*Good catch! Added this detail.*

**Figure 12: (a) Also for the LISTOS comparison only the extended stations are used**
*I clarified this figure caption now to ensure it states that it is only the four stations with extended temporal extent.*

**Supplement Line 53: "to remove the estimated of loss of sensitivity"**
**First "of" can be removed**
*This phrase was corrected.*

---

## Author Comment (AC3) · 15 Aug 2020

*Reviewer 2:*
*We would like to thank you for your comments as we appreciate the time dedicated for this review and have made changes to the manuscript to reflect the suggestions. Individual comments from the review are bolded below with our responses in italics.*

**The present manuscript presents the evaluation of S5P TROPOMI tropospheric NO2 column densities with the aid of airborne and ground-based spectrometers in New York City and Long Island Sound. The advantage/ challenge of this region is that the NO2 concentrations are highly heterogeneous in time and space. The validation of S5P TROPOMI tropospheric NO2 column densities is separated in two major categories: (1) comparison between airborne NO2 TrVC and TROPOMI NO2 TrVC and (2) comparison between ground-based NO2 TrVC and TROPOMI NO2 TrVC. From the abovementioned comparisons, the authors observe a bias in TROPOMI NO2 TrVC and the effect of clouds and a-priori profile in the TROPOMI retrieval are examined into details. I strongly recommend the publication of the manuscript after consideration of a minor number of specific considerations:**

**Specific comments:**
**– Page 2, Line 60: It would be interesting to add the exact spatial resolution of OMI and OMPS.**
*Spatial resolutions were added to this discussion.*

**– Page 4, Line 110: I suggest that for the reader it would be more practical if you include a small separate section or subsection called "LISTOS campaign" and write there the information about the campaign, as you already did in Section 2.**
*The first paragraph in Section 2 serves this purpose as the campaign description. I added a subsection header to this paragraph so that a reader can easily identify where the campaign information is discussed.*

**– Page 7, Line 218: Please explain the PRATMO acronym**
*In other sources that cite this model, PRATMO is not defined as an acronym. However, we did find out it is short for 'Prather Atmospheric Model'. I defined this in the text.*

**– Page 8, Line 234: If I understand well, did you assume that the aerosol a-priori profile in the AMF calculation is zero? So, you assumed that no aerosols are present in the atmosphere, or not? If this the case, is this assumption leading to realistic results?**
*You understood the assumption correctly that aerosol a priori profile is zero for this analysis. However, as shown by our comparison to Pandora, in which the direct-sun measurements are largely insensitive to aerosols at the levels observed, we still compare really well so we expect impacts due to aerosols to be smaller than the other sources of bias we have found in this analysis. However, in future work, we plan to incorporate the HALO aerosol profile data into our retrieval to directly assess potential impacts.*

**– Page 18, Line 580: Can you provide an approximate value for the Pandora horizontal sensitivity?**
*Assuming this comment is referring to the horizontal bars in the figure mentioned (Figure 10), the temporal variation in Pandora is proportional to pollution level ($10^{th}$-$90^{th}$ percentile range vs. Pandora TrVC: $r^2$=0.69 and y(range)=0.47(TrVC)-0.52x$10^{15}$ molecules cm$^{-2}$). I added the following sentence to the text: 'Pandora's temporal variation is proportional to pollution level ($r^2$=0.69).'*

**– Page 19, Line 581: This means that the Pandora data are not filtered for clouds?**
*I think you are referring to the phrase 'Although cloud information for Pandora comparisons at TROPOMI sub-pixel resolution is not readily available…' does make it sound like Pandora data were not cloud filtered but this is not the case. Clouds are filtered in Pandora algorithm through their quality flags. The intention was that Pandora has the ability to still have a clear direct line of the sun even if TROPOMI has broken clouds (elevated*

*cloud fractions), but we don't have direct measurements of sub-pixel cloud coverage like we do from the airborne spectrometer data.*

*I changed this sentence to be clearer: 'Unlike with airborne spectrometer data, sub-TROPOMI pixel cloud information is not readily available for these comparisons to Pandora. However, the impact of coincidence criteria…'*

**– Page 21, Line 650: Is there a reason why you did not compare Pandora TrVC (vs) TROPOMI-NAMCMAQ for the extended time period? I would be interesting to add a figure with this comparison.**
*The NAMCMAQ runs that we used are only available through September 2018 as they were run as part of the LISTOS campaign and not operationally. Future work in this area (as many of these Pandoras will operate over a longer time period in this region) will consider sources for higher resolution a priori data available over a longer time period (e.g., the NASA GEOS-CF at 25 km resolution) and assess the impact to the results.*

**– Page 21, Line 678: You could cite studies that use MAX-DOAS measurements to evaluate the TROPOMI NO2 product.**
*I added studies using the MAX-DOAS technique to evaluate TROPOMI NO2. Though it is noted 1 has been accepted but not available yet (Chan), and the final two are still in discussion in AMTD. .*

*Chan, K. L., Wiegner, M., Alberti, C., and Wenig, M.: MAX-DOAS measurements of tropospheric NO₂ and HCHO in Munich and the comparison to OMI and TROPOMI satellite observations, Atmos. Meas. Tech. Discuss., https://doi.org/10.5194/amt-2020-35, in review, 2020.*

*Liu, M., Lin, J., Kong, H., Boersma, K. F., Eskes, H., Kanaya, Y., He, Q., Tian, X., Qin, K., Xie, P., Spurr, R., Ni, R., Yan, Y., Weng, H., and Wang, J.: A new TROPOMI product for tropospheric NO₂ columns over East Asia with explicit aerosol corrections, Atmos. Meas. Tech., 13, 4247–4259, https://doi.org/10.5194/amt-13-4247-2020, 2020.*

*Dimitropoulou, E., Hendrick, F., Pinardi, G., Friedrich, M. M., Merlaud, A., Tack, F., De Longueville, H., Fayt, C., Hermans, C., Laffineur, Q., Fierens, F., and Van Roozendael, M.: Validation of TROPOMI tropospheric NO₂ columns using dual-scan MAX-DOAS measurements in Uccle, Brussels, Atmos. Meas. Tech. Discuss., https://doi.org/10.5194/amt-2020-33, in review, 2020.*

*Verhoelst, T., Compernolle, S., Pinardi, G., Lambert, J.-C., Eskes, H. J., Eichmann, K.-U., Fjæraa, A. M., Granville, J., Niemeijer, S., Cede, A., Tiefengraber, M., Hendrick, F., Pazmiño, A., Bais, A., Bazureau, A., Boersma, K. F., Bognar, K., Dehn, A., Donner, S., Elokhov, A., Gebetsberger, M., Goutail, F., Grutter de la Mora, M., Gruzdev, A., Gratsea, M., Hansen, G. H., Irie, H., Jepsen, N., Kanaya, Y., Karagkiozidis, D., Kivi, R., Kreher, K., Levelt, P. F., Liu, C., Müller, M., Navarro Comas, M., Piters, A. J. M., Pommereau, J.-P., Portafaix, T., Puentedura, O., Querel, R., Remmers, J., Richter, A., Rimmer, J., Rivera Cárdenas, C., Saavedra de Miguel, L., Sinyakov, V. P., Strong, K., Van Roozendael, M., Veefkind, J. P., Wagner, T., Wittrock, F., Yela González, M., and Zehner, C.: Ground-based validation of the Copernicus Sentinel-5p TROPOMI NO₂ measurements with the NDACC ZSL-DOAS, MAX-DOAS and Pandonia global networks, Atmos. Meas. Tech. Discuss., https://doi.org/10.5194/amt-2020-119, in review, 2020.*

**– Page 22, Line 699: Please add some reference studies.**
*A couple references were added to the text in reference to this line.*

**– Page 39, Table 5: Is there a reason why you did not present the median percentage difference for the Standard Slant Column?**
*I may have had a reason initially, which was likely because it was a slant column comparison and we don't expect them to be comparable, however, I added the statistics just to be consistent since I do report the linear fits (which demonstrates how correlated the two datasets are).*

**– Page 46, Figure 6: I would suggest that in Fig. 6a, you include the reported TROPOMI SCD error.**

*There is not a reported tropospheric SCD precision in the product file. However, I did look into calculating an uncertainty for tropospheric slant column by considering the slant column equivalent of the first two terms in equation 22 in the ATBD (*[http://www.tropomi.eu/sites/default/files/files/publicS5P-KNMI-L2-0005-RP-ATBD_NO2_data_products-20190206_v140.pdf](http://www.tropomi.eu/sites/default/files/files/publicS5P-KNMI-L2-0005-RP-ATBD_NO2_data_products-20190206_v140.pdf)*) and the value is small enough that the vertical error bars are not visible in Figure 6a. The mean is $5.5 \times 10^{14}$ molecules cm$^{-2}$ with a standard deviation of $7.4 \times 10^{13}$ molecules cm$^{-2}$. In the figure caption, I added a statement about this.*

**– Page 47, Figure 7: The figure does not contain error bars in the vertical axis. Is there any way to estimate the TROPOMI-NAMCMAQ error and add it to the figure?**
*We do not have an estimate of TROPOMI-NAMCMAQ error.*

---

## Author Comment (AC4) · 16 Aug 2020

Upon creating my final response, I realized I answered one of my responses in reference to the wrong line in the manuscript and therefore changes the context of the question. It is in reference to the comment below where I addressed it better here.

– Page 18, Line 580: Can you provide an approximate value for the Pandora horizontal sensitivity? Pandora's approximate horizontal path through the lower troposphere, where most of the NO2 resides, is quite small and local (less than 1 km). For example,

[Figure]

if we had a 1km mixed layer height and the range of SZAs during the LISTOS time period (17-40 degrees), the horizontal path length through the mixed layer would range from ∼300-840 m.

---

## Author Response (AR2)

***Thank you for your minor comments to this manuscript. I am happy to make the changes as suggested. I commented on each of your comments below in bold-italics and the tracked changes are found underneath.***

1. There is a special issue on TROPOMI (joint AMT/ACP):
https://amt.copernicus.org/articles/special_issue1002.html
You might consider asking Copernicus whether your study could be added to this special issue in order to increase visibility; I don't know if it is still possible, but it might be worth trying.

***I had initially missed this deadline for that issue due to a family emergency taking me away from work for a few weeks but had put a note in the cover letter to see if an exception could have been made but I didn't hear anything about that option. Any idea on how to ask for an exception through a different channel? Is this something as editors you could help with?***

2. The TROPOMI NO2 algorithms have been improved over the last months. It will take some time until all data can be reprocessed, but it might be very interesting to check the comparisons of the reprocessed data with the existing airborne/Pandora measurements in future.
I hope that your dataflow is largely automatized so that such an update would be possible without much effort.
It might be worth mentioning this aspect as outlook in the conclusions.
***Definitely! I have it set up where it would be relatively simple to switch out to a new reprocessed dataset and plan to do that when it is available. I had already reflected on this in the final paragraph. I did some revisions on that paragraph though and it now reads:***

> ***As the spatial and temporal resolution of satellite-based observations have and will continue to improve in the near future, gathering large datasets of coincident observations with airborne spectrometers become more feasible during air quality field studies. This provides a unique perspective for satellite validation and evaluation strategies, especially with the added information on sub-pixel variability compared to traditional reference datasets. The datasets presented in this work and others like it will continue to provide a reference for validating and evaluating UV-VIS trace gas retrievals, including the assessment of reprocessed TROPOMI products and near-future geostationary measurements.***

3. As far as I understood, the problem with CTH occurs where the assumend surface albedo is wrong (as a consequence of the coarse resolution of the albedo map).
I would appreciate if this aspect could be specified when mentioning the CTH issue in abstract and conclusions.
CTH meaning (cloud top height?) or something like that. To make sure I understand this correctly
***Good suggestion. I put some text in the conclusions and abstract to reflect this. In the abstract, text reads: 'The largest outliers between TROPOMI and the reference measurements stem from too spatially coarse a priori surface reflectivity (0.5°) over bright urban scenes. In***

[revised manuscript text omitted]